# CRAFTING ZERO-COST PROXY METRICS FOR NEURAL ARCHITECTURE SEARCH VIA SYMBOLIC REGRESSION

## ABSTRACT

Using zero-cost (ZC) metrics to estimate network performance instead of relying on expensive training processes has proven both its efficiency and efficacy in Neural Architecture Search (NAS). However, a significant limitation of most ZC proxies is their inconsistency, as reflected by the substantial variation in their performance across different problems. Additionally, the design of current ZC metrics is manual, which is a lengthy trial-and-error process and requires expert knowledge to develop ZC metrics effectively. These challenges raise two questions: **(1) Can we automate the design of ZC metrics?** and **(2) Can we utilize the existing hand-crafted ZC metrics to synthesize a better one?** In this study, we propose a framework based on Symbolic Regression to automate the design of ZC metrics. Our framework is not only highly extensible but also capable of quickly producing a ZC metric with a strong positive rank correlation to network performance across multiple problems within just a few minutes. Extensive experiments on 13 problems in NAS-Bench-Suite-Zero, covering various search spaces and tasks, demonstrate the superiority of our automatically designed proxies over hand-crafted ones. By integrating our proxy metrics into an evolutionary algorithm, we could identify a network architecture with comparable performance on the CIFAR-10 dataset within 15 minutes using a single GeForce RTX 3090 GPU[1].

## 1 INTRODUCTION

Neural architecture search (NAS), a subfield of Automated Machine Learning (AutoML), focuses on the automatic design of high-performance network architectures (Elsken et al., 2019). NAS can be described as a procedure in which a *search algorithm* explores a *search space* containing potential architectures. During the exploration, the search algorithm assesses the quality of candidate networks using a *performance estimation strategy*. Early NAS studies estimated the quality of networks based on their performance on the validation dataset (Zoph & Le, 2017; Real et al., 2019). Although top-performing networks could be obtained, a single NAS run took several weeks or months to complete and heavily depended on hardware resource (i.e., Real et al. (2019)

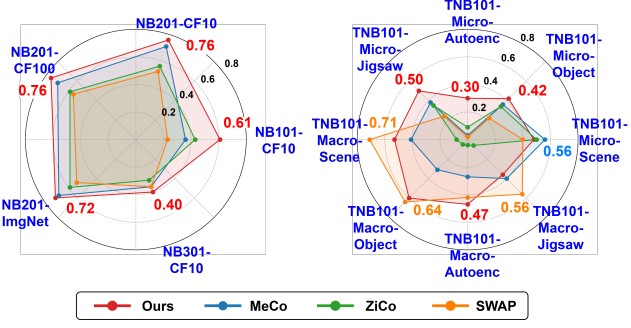

Figure 1: Comparisons in Kendall's Tau rank correlations between our automatically-designed ZC metric and state-of-the-art ZC metrics (i.e., MeCo, ZiCo, and SWAP) on NAS-Bench-101/201/301 (Left) and TransNAS-Bench-101-Micro/Macro (Right). Highest Kendall's $\tau$ scores for each problem are presented. The results highlight the consistency of our ZC metric across various search spaces and tasks.

---

[1]Our source code is provided in the supplementary material.

used 800 GPUs for a single run). Recent works have proposed various approaches to deal with this bottleneck such as weight-sharing (Pham et al., 2018), performance prediction (White et al., 2021), or zero-cost metrics (Abdelfattah et al., 2021). Among these techniques, using *zero-cost* metrics (also known as *training-free* metrics in the literature) offers the most efficiency since it can shorten the search time from days to a dozen seconds or minutes. For example, Peng et al. (2024) found a network architecture with competitive performance on CIFAR-10 within 6 minutes using SWAP, which is a zero-cost proxy that measures the expressivity of the network. However, a significant limitation of the zero-cost metrics (ZC metrics) is their *order-preserving ability*. The order-preserving ability of ZC metrics can be assessed by examining the consistency between the rankings of architectures produced by ZC proxies and those based on their true performance (Zhang et al., 2024). Many studies have highlighted inconsistencies in network rankings based on ZC metrics' scores across different problems (Abdelfattah et al., 2021; Krishnakumar et al., 2022). While ZC metrics exhibit positive rank correlations with network performance on some problems, they may demonstrate negative correlations on others. Recent studies have attempted to address this issue when newly proposed ZC proxies (e.g., ZiCo, MeCo, SWAP) achieve positive rank correlations across all testing problems (Li et al., 2023; Jiang et al., 2023; Peng et al., 2024). In these cases, a network that receives a high ZC score is also likely to exhibit high performance, and *vice versa*. However, despite achieving positive rank correlations, the scores for some problems remain close to 0.0, indicating a lack of correlation between the ZC scores and the networks' performance. In this study, our target is to design a ZC metric that yields not only **the consistent order-preserving ability** but also **a high positive rank correlation** across various problems[2].

Table 1: List of frameworks that automatically design ZC metrics.

| Name | Method | Features | Multiple-problems | High-extensibility |
|------|--------|----------|------------------|--------------------|
| EZNAS (Akhauri et al., 2022) | Symbolic Regression | Low-level | ✗ | ✗ |
| Auto-Prox (Wei et al., 2024) | Symbolic Regression | Low-level | ✔ | ✗ |
| UP-NAS (Huang et al., 2024) | Neural network | High-level | ✗ | ✔ |
| **Ours** | Symbolic Regression | High-level | ✔ | ✔ |

Most proposed ZC metrics are manually designed with expert knowledge. *Can we automate the design of efficient ZC metrics to form the proxies that are better than hand-crafted ones?* The practicability of this approach has been demonstrated in several studies. EZNAS (Akhauri et al., 2022) implemented Symbolic Regression to form a ZC metric by combining three low-level features of networks: weights, activations, and gradients. While EZNAS only considered convolutional neural networks, Auto-Prox (Wei et al., 2024) extended the framework to design ZC metrics for vision transformer architectures. Additionally, a single run of Auto-Prox aims to search for a ZC metric that could perform well across multiple problems instead of a single problem as in EZNAS. UP-NAS (Huang et al., 2024) designed a new ZC metric by combining higher-level features of networks (i.e., hand-crafted ZC metrics) and taking their weighted sum, in which the weights of metrics are found by using a neural network. However, a search process is needed to obtain suitable weights whenever we handle a new problem. In this paper, we focus on the automatic designing of efficient ZC metrics. Table 1 highlights the differences between our proposed framework and previous approaches that also automate the design of ZC metrics. We discuss in detail the differences in Section 3. Generally, our framework is designed to synthesize a new ZC metric from existing hand-crafted ones, and the outcome metric could perform well across diverse problems. Our method offers high extensibility, making it straightforward to adapt to various problems or modify input features to enhance the performance of the resulting metric. Our **main contributions** are as follows:

- We propose a highly extensible framework for automatically designing a ZC metric that not only consistently preserves the order of network performance but also achieves a high positive rank correlation across multiple problems.

---

[2]In this paper, we define an NAS problem consisting of a single search space (e.g., NAS-Bench-101, DARTS) and a single task (e.g., image classification on CIFAR-10 dataset, image classification on ImageNet dataset, object detection on Taskonomy dataset). If we search network architectures within one search space but evaluate them on two different tasks, we consider that we are solving two NAS problems. We list all 13 NAS problems in our experiments in Appendix C.

- Our framework can synthesize a new ZC metric within only 10 minutes, which is significantly faster than the 24 hours required by EZNAS. Compared to previous ZC metrics, including newly proposed ones (i.e., ZiCo, MeCo, SWAP), our metric achieves the state-of-the-art Kendall's tau correlation for NAS-Bench-101/201/301 search spaces and comparable scores for TransNAS-Bench-101-Micro/Macro search spaces (see Figure 1).

- The performance of the best networks found by our synthesized ZC proxy surpasses those found by hand-crafted ZC proxies in 8 out of 13 problems in NAS-Bench-Suite-Zero. By integrating the found ZC metric with a simple evolutionary algorithm, we could find a network with comparable performance on the CIFAR-10 dataset within 15 minutes, demonstrating the practical applicability of our obtained ZC metric in real-world NAS scenarios.

## 2 RELATED WORK

**Limitation of ZC-NAS metrics**   Zero-cost metrics are widely employed in NAS due to their ability to estimate the network performance with a trivial cost (Krishnakumar et al., 2022). However, several studies have indicated their inconsistent order-preserving ability across different problems (Abdelfattah et al., 2021; Krishnakumar et al., 2022). For example, the ZC metric **Grad-norm** has the Spearman rank correlation to network performance of $0.58$ for the NAS-Bench-201 search space but has a score of $-0.21$ for the NAS-Bench-NLP search space (Abdelfattah et al., 2021). Recent proposed ZC metrics (e.g., **ZiCo**, **MeCo**, **SWAP**) have tackled this issue since they consistently yield a positive rank correlation across various problems. Nevertheless, their rank correlation scores are close to $0.0$ for some cases (e.g., the **ZiCo** metric for the TransNAS-Bench-101-Macro search space), exhibiting the lack of correlation between the ZC metrics' scores and the networks' performance. Designing a ZC metric that exhibits a high correlation to networks' performance and a strong order-preserving ability across various problems is essential.

**Automatic designing of ZC-NAS metrics**   While most ZC metrics are manually designed with expert knowledge, several studies aim to design ZC metrics *automatically* (Akhauri et al., 2022; Wei et al., 2024; Huang et al., 2024). **EZNAS** (Akhauri et al., 2022) proposed a framework that uses Symbolic Regression (SR) to synthesize a ZC metric from three basic statistics of convolutional neural networks: weights, activations, and gradients. The empirical results shown that the ZC metric found by EZNAS could surpass hand-crafted metrics. **Auto-Prox** (Wei et al., 2024) then modified the framework of EZNAS to find a ZC metric for vision transformer networks. In EZNAS, the dataset used by SR consists of networks' information in a single problem. Such an approach might make the resulting ZC metric overfitted on the experimented problem and perform worse on the unseen problems. Auto-Prox solved this issue by combining multiple problems into the dataset of SR instead of a single one. However, Auto-Prox only tested the proposed framework in finding the ZC proxy for candidate networks within the same search space. When facing a different search space, the search process was re-conducted to synthesize a new ZC proxy. On the other hand, both EZNAS and Auto-Prox are restricted from discovering more robust ZC metrics when they synthesize new ZC proxies using *low-level* features of the network. Compared to EZNAS and Auto-Prox, the extensibility of **UP-NAS** (Huang et al., 2024) is higher when it synthesized new ZC metrics from *high-level* features: hand-crafted ZC metrics. UP-NAS assumed there is a linear relationship between hand-crafted ZC metrics and used a neural network to find the weights of 13 ZC metrics listed in NAS-Bench-Suite-Zero. A new ZC proxy was then formed by taking their weighted sum. However, the limitation of UP-NAS is similar to EZNAS as its dataset only covers a single problem. A new ZC metric thus needs to be searched for whenever handling a different NAS problem.

**NAS Benchmarks**   An NAS benchmark can be viewed as a database consisting of essential information about candidate networks (e.g., train/validation/test performance, the number of parameters) within the same search space. The **NAS-Bench-101/201/301** benchmarks (Ying et al., 2019; Dong & Yang, 2020; Siems et al., 2020) provide the performance of candidate networks for the image classification task with different datasets. The **TransNAS-Bench-101-Micro/Macro** benchmarks (Duan et al., 2021) consider the network performance on other tasks beyond the image classification such as scene classification, and object detection. The **NAS-Bench-Suite-Zero** benchmark (Krishnakumar et al., 2022) is the collection of the aforementioned benchmarks and additionally computes the scores for all candidate networks in terms of 13 various ZC metrics. The clear advantage of using NAS benchmarks is that we can quickly evaluate the effectiveness of our methods and fairly

compare them to others in the same setting. In this paper, besides using NAS benchmarks to validate our proposed method, we utilize NAS benchmarks to create the dataset for our framework to automatically learn new ZC metrics.

# 3 PROPOSED FRAMEWORK

Our proposed framework for automating the design of ZC-NAS metrics is based on Symbolic Regression (SR). Given existing hand-crafted ZC metrics, we use SR to search for the most effective way to combine these metrics using mathematical operators (e.g., *add*, *mul*). In this section, we first outline our approach to building the dataset for putting into SR in Section 3.1. The proposed mechanism for evaluating the quality of ZC metrics during the search process is then detailed in Section 3.2. The method for representing the synthesized ZC proxies and the ways that SR discovers novel ZC proxies are presented in Sections 3.3 and 3.4, respectively. The entire search procedure of our framework is discussed in Section 3.5. While the use of SR for automatically designing ZC metrics was introduced in EZNAS (Akhauri et al., 2022) and Auto-Prox (Wei et al., 2024), there are notable differences between our framework and theirs, particularly in the construction of the input dataset for SR, the mechanism for evaluating synthesized ZC metrics, and the search procedure. These differences are thoroughly discussed in the following sections.

## 3.1 DATASET BUILDING

Our approach to building the input dataset for SR closely resembles that of UP-NAS (Huang et al., 2024), which uses hand-crafted ZC-NAS metrics as feature variables and validation performance of networks as the ground truth. However, while the dataset of UP-NAS only covers a single problem, our dataset encompasses the networks' information across *multiple problems*. Although the suggestion of integrating multiple problems into the input dataset of SR was proposed in Auto-Prox (Wei et al., 2024), Auto-Prox uses the low-level features (i.e., network statistics), whereas we utilize the high-level features (i.e., hand-crafted ZC metrics).

When creating the input dataset, while the cost of computing the ZC proxy values is negligible, the most significant computational cost arises from defining the ground truth (i.e., the network performance). However, we overcome this obstacle by employing existing NAS benchmarks (e.g., NAS-Bench-101, NAS-Bench-201), which provide the performance of numerous candidate architectures across various search spaces and tasks. Using NAS benchmarks also demonstrates strong extensibility. Whenever a new NAS benchmark or hand-crafted ZC metric is introduced, we can quickly compute the ZC metric scores of networks at a negligible cost and enrich the input dataset by integrating the new information. Another way of defining the ground truth involves using supernets (e.g., Once-for-All (Cai et al., 2020)) or performance predictors (e.g., XGBoost (White et al., 2021)), for which configurations and weights are available. In this study, we primarily focus on NAS benchmarks to utilize truly trained performance rather than predicted performance.

## 3.2 OBJECTIVE EVALUATION

Our target is to design a ZC metric that ranks candidate networks in the same order as their true performance. We thus use Kendall's $\tau$ rank correlation to evaluate the quality of ZC metrics generated by SR during the search process, similarly to EZNAS, UP-NAS, and Auto-Prox. However, the evaluation methods used in EZNAS and UP-NAS are not applicable to our framework, as our dataset comprises multiple problems rather than a single one. Auto-Prox, which also uses multiple problems to build the input dataset, evaluates the quality of an arbitrary ZC metric by computing its Kendall's $\tau$ value for each problem and taking the weighted sum. This approach has its limitation. In particular, because the weight for each problem must be *pre-defined* before searching, it may cause the resulting ZC metric to be biased towards the problem with the highest weight. Setting equal weights for all problems is not effective due to the differences in Kendall's $\tau$ values across problems, leading SR to focus disproportionately on the problem that produces the highest rank correlation and neglect others.

Our proposed mechanism for evaluating ZC metrics generated by SR is as follows. Given $\{\tau_1, \tau_2, \ldots, \tau_N\}$ as the set of Kendall's $\tau$ values of an arbitrary ZC metric $\boldsymbol{x}$ on $N$ problems, the

quality of $\boldsymbol{x}$ (denoted as $Score(\boldsymbol{x})$) is defined as:

$$Score(\boldsymbol{x}) = \sum_{i=1}^{N} \frac{\tau_i - \tau_i^-}{\tau_i^+ - \tau_i^-}, \tag{1}$$

where $\tau_i^-$ and $\tau_i^+$ represent the lowest and highest Kendall's $\tau$ values obtained so far for the $i$-th problem. Equation 1 scales Kendall's $\tau$ score for each problem into the range $[0,1]$, with the new value being 0 and 1 if it corresponds to the lowest and highest scores obtained so far for the problem, respectively. The maximum score for the ZC metric $\boldsymbol{x}$ is therefore $N$, if it yields the highest Kendall's $\tau$ values for all problems at the time of evaluation.

### 3.3 SOLUTION REPRESENTATION

Our use of SR to design the ZC metric is similar to finding the optimal function that combines multiple hand-crafted ZC metrics, which can be represented as an expression tree. Figure 2 illustrates an example of a function combining two ZC proxies and the corresponding expression tree. In an expression tree, the *leaf* nodes represents variables or constants (e.g., nodes '**Snip**' and '**MeCo**' in Figure 2), while the *internal* nodes represents mathematical operators (e.g., nodes '$+$' and '$\times$' in Figure 2). In our experiments, we constrain the search space by using a fixed set of primitive mathematical operators $\{add,$ $sub, mul, div, neg, log, sqrt\}$ (more details on

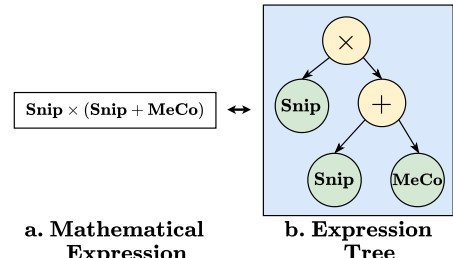

**a. Mathematical Expression**     **b. Expression Tree**

Figure 2: Example of using expression tree to represent the expression $\mathbf{Snip} \times (\mathbf{Snip} + \mathbf{MeCo})$ in SR.

these operators are provided in Appendix B), and we exclude constants in the mathematical expressions, meaning that the leaf nodes are exclusively ZC metrics. The minimum and maximum tree depths are also restricted to 2 and 10, respectively.

### 3.4 VARIATION OPERATORS

We implement basic variation operators, including crossover, subtree mutation, hoist mutation, and point mutation, to enable SR to discover novel expression trees (i.e., ZC metrics) during the search process. Illustrations of these operators are exhibited in Figure 3.

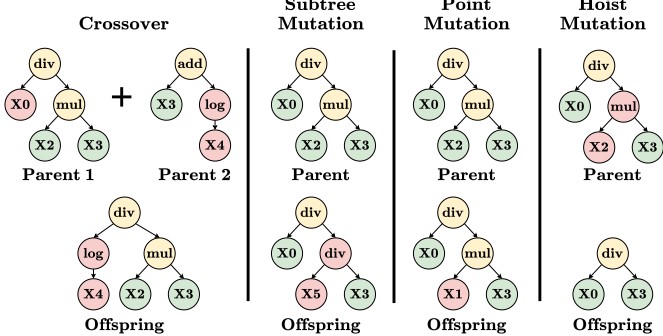

Figure 3: Illustration of crossover and mutation operators for expression trees.

### 3.5 SEARCH PROCEDURE

Figure 4 illustrates the entire procedure we use SR for synthesizing new ZC metrics in this study. Initially, SR generates a population of $N$ synthesized ZC metrics, which are represented by expression trees as described in Section 3.3. We then evaluate the quality of these ZC metrics using the

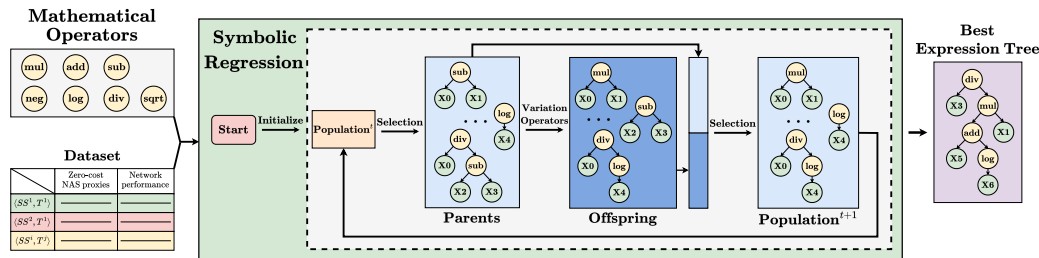

Figure 4: The procedure of searching new ZC metrics using Symbolic Regression. Our dataset consists of multiple NAS problems and it thus contains multiple search spaces $SS = \{SS_1, SS_2, \ldots, SS_i\}$ and tasks $T = \{T_1, T_2, \ldots, T_i\}$. $SS_i$ refers to the $i$-th search space in the list of search spaces $SS$ and $T_j$ refers to the $j$-th in the list of tasks $T$.

method detailed in Section 3.2 and employ binary tournament selection to choose parents for generating offspring expressions. Given a set $S$ of $N$ individuals, we divide $S$ into $N/2$ pairs and select the winner from each pair based on their quality. This process is repeated until $N$ parent expressions (i.e., winners) are selected. The parent expressions then undergo the variation operators (i.e., crossover and mutation, as discussed in Section 3.4) to produce offspring expressions. The offspring are evaluated for their quality, and along with the parents, form a set $P+O$ with a size of $2 \times N$. We again perform tournament selection on the set $P + O$ to select the $N$ best expression trees for the population in the next generation. The search process of SR is executed until the stopping criteria are satisfied (e.g., reaching the maximum number of generations), and the final output of SR is the best expression tree in the final population.

We highlight the differences between our search procedure and those used in EZNAS and Auto-Prox. In their frameworks, after the offspring are produced and evaluated for quality, they are immediately chosen as individuals for the next population, without comparing to the parents as in our approach. The drawback of such non-elitist approaches is that good solutions could be inadvertently eliminated and replaced with inferior ones. In contrast, by comparing parents and offspring, we ensure that high-quality ZC metrics remain in the population until the end of the search process.

## 4    EXPERIMENTS AND RESULTS

The majority of our experiments are conducted on **NAS-Bench-Suite-Zero** (Krishnakumar et al., 2022), which consists of five different NAS benchmarks: **NAS-Bench-101** (Ying et al., 2019), **NAS-Bench-201** (Dong & Yang, 2020), **NAS-Bench-301** (Siems et al., 2020), and **TransNAS-Bench-101-Micro/Macro** (Duan et al., 2021)). The network architectures in **NAS-Bench-101/201/301** are evaluated for the **Image classification** task, while those in **TransNAS-Bench-101-Micro/Macro** are evaluated on the tasks of **Object detection**, **Scene classification**, **Jigsaw puzzle**, and **Autoencoding**[3]. In summary, we experiment and validate our proposed method on **13** problems in NAS-Bench-Suite-Zero (more details of each problem are provided in Appendix C). Beyond NAS benchmarks, we also demonstrate the effectiveness of our SR-designed ZC metric in practice by testing it on large-scale search spaces (i.e., DARTS (Liu et al., 2019), Once-For-All (Cai et al., 2020)) and dataset (i.e., ImageNet). All experiments are conducted on a single GeForce RTX 3090 GPU.

### 4.1    SEARCHING FOR A ROBUST ZC METRIC ACROSS MULTIPLE PROBLEMS

We first run the proposed framework with the input dataset containing the values of 16 hand-crafted ZC-NAS proxies and the validation accuracies of network architectures across *three* search spaces: NAS-Bench-101, NAS-Bench-201, and NAS-Bench-301. Both the ZC scores and validation accuracies are measured on the CIFAR-10 dataset. For ZC proxies, we reuse the values of 13 ZC metrics that are logged in NAS-Bench-Suite-Zero, including **FLOPs**, **Params**, **Jacov** (Mellor et al., 2020), **NWOT** (Mellor et al., 2021), **Synflow** (Tanaka et al., 2020), **Snip** (Lee et al.,

---

[3]There are three additional tasks **Room Layout**, **Surface Normal** and **Semantic Segmentation** for the two TransNAS-Bench-101 search spaces, but we encountered a dataset issue similar to Peng et al. (2024).

2019), **EPE-NAS** (Lopes et al., 2021), **Fisher** (Turner et al., 2020), **Grad-norm** (Abdelfattah et al., 2021), **Grasp** (Wang et al., 2020), **L2-norm** (Abdelfattah et al., 2021), **Zen** (Lin et al., 2021), **Plain** (Abdelfattah et al., 2021)). Additionally, we compute the values of three state-of-the-art ZC metrics by using their published source code: **ZiCo** (Li et al., 2023), **MeCo** (Jiang et al., 2023), and **SWAP** (Peng et al., 2024). For each search space, we add 70% of the total networks into the input dataset for SR and use the remaining 30% for the test dataset. We run and test our framework for 31 independent runs with different seeds. The population size of SR is set to 100 and the search process is terminated when SR reaches 50 generations. Consequently, we explore a total of 5,000 synthesized ZC metrics at each run. Additional hyperparameters for variation operators (e.g., the crossover and mutation probabilities) are presented in Appendix D.

The obtained results reveal differences in the formulas of the ZC metrics designed by our framework across 31 independent runs (see Appendix H). However, the quality of these expressions is relatively consistent, as indicated by the small standard deviations (i.e., approximately 0.01 to 0.03) over 31 runs. The results also show the similarity in the rank correlation of SR-designed ZC metrics on the input and test datasets, suggesting that the designed metrics are not overfitted to the input data and perform well across the entire dataset. We also observe the differences in Kendall's $\tau$ scores achieved across different problems. While we can obtain an average score of 0.74 for the NB201-CF10 problem, the scores for NB101-CF10 and NB301-CF10 problems are approximately 0.56 and 0.38, respectively. Nonetheless, it is noteworthy that our $\tau$ values are significantly higher than those achieved by hand-crafted ZC proxies for these problems (comparisons are provided in Section 4.2).

We further analyze the ZC metrics designed by our framework. As shown in Figure 5, each hand-crafted ZC metric is used at least once in the synthesized metrics. Notably, the **MeCo** metric is selected by SR in all 31 runs, whereas the **SWAP** metric is only chosen once. Although **SWAP** exhibits relatively high rank-correlation across the three experimental problems (see Figure 6), its combination with other hand-crafted ZC metrics appears to be less promising. Interestingly, **Snip** is the second most chosen metric by our framework despite its modest $\tau$ scores across the three experimental problems. These findings suggest that our framework does not merely favor metrics with high rank-correlation but instead prioritizes those metrics that contribute most effectively when combined with others.

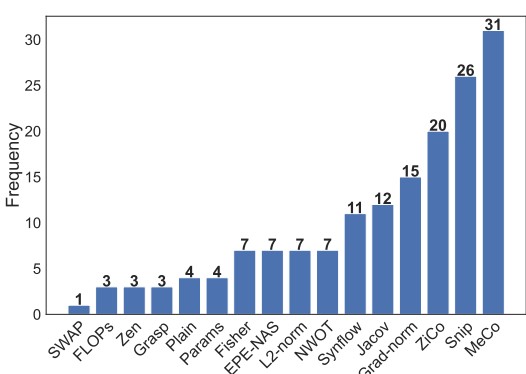

Figure 5: Frequency of hand-crafted ZC metrics in the 31 final ZC metrics synthesized by SR.

Among 31 final ZC metrics returned by our framework, we evaluate Kendall's $\tau$ for each problem and compute the score for each candidate according to Equation 1. The metric with the highest score is selected for use and comparison with other ZC metrics in later experiments. This ZC metric is presented in Equation 2, which combines six hand-crafted ZC metrics: **FLOPs**, **Snip**, **L2-norm**, **Zen**, **ZiCo**, and **MeCo**.

$$ f(\cdot) = \frac{ZiCo \times MeCo^2 \times log(FLOPs)}{(MeCo + Zen) \times (\sqrt{Snip} \times (MeCo + Zen + 2 \times \text{L2-norm}) + MeCo)} \tag{2} $$

## 4.2 COMPARISON TO OTHER ZC METRICS ON NAS-BENCH-SUITE-ZERO

**Comparison to hand-crafted ZC metrics:** Figure 6 exhibits the Kendall's $\tau$ score comparisons between our obtained ZC metric (i.e., Equation 2) (denoted as **SR-NAS**) and hand-crafted ZC metrics across 13 problems in NAS-Bench-Suite-Zero. We additionally visualize the correlation between our ZC metric and the ground-truth performance in Appendix G. The results indicate that Kendall's $\tau$ scores of our ZC metric significantly surpass those of other metrics for the problems used to build the input dataset of our framework (i.e., NB101-CF10, NB201-CF10, and NB301-CF10). For the NB201-CF100 and NB201-IMGNT problems, which reside within the same search space as one of the problems in the input dataset (i.e., NB201-CF10), our ZC metric also achieves the highest Kendall's $\tau$ scores. We further assess the generalizability of our metric by validating it on unseen

Figure 6: Kendall's $\tau$ rank correlation between ZC metrics values and networks' ground truth performance. The rows are arranged based on the average rank of metrics across all problems, with problems within the same search space colored identically. The highest Kendall's $\tau$ for each problem is highlighted with a red box. The results are presented in three groups: **(a)** Problems *included* in the dataset used by our framework in the search procedure; **(b)** Problems within the same search space as one of the problems in (a), but with different tasks; **(c)** Problems *not included* in the dataset used by our framework in the search procedure. Our ZC metric is denoted as **SR-NAS**.

search spaces (i.e., TransNAS-Bench-101 Micro/Macro) and tasks (i.e., Object detection, Scene classification, Jigsaw puzzle, and Autoencoding). As depicted in Figure 6, our metric yields the highest scores in 3 out of 4 problems in the TransNAS-Bench-101-Micro and achieves comparable scores for the remaining problems. Additionally, we rank the ZC metrics based on the sum of their rankings across all problems. Figure 6 shows that **SR-NAS** holds the top rank, underscoring the impressive stability of our ZC metric across various problems. These results demonstrate the effectiveness of leveraging hand-crafted ZC metrics to automatically synthesize a robust ZC metric that performs consistently well across multiple search spaces and tasks.

**Comparison to other automatic frameworks:** Table 2 presents a comparison of both Kendall's $\tau$

Table 2: Comparisons in Kendall's $\tau$ rank correlation between frameworks that automate the design of ZC metrics. The symbol '-' indicates that the results were not reported in the original studies. The best results are presented in red color. We do not compare to Auto-Prox (Wei et al., 2024) because it serves for vision transformer networks and does not test on NAS-Bench-Suite-Zero.

| Framework | NB201-CF10 | NB201-CF100 | NB201-IMGNT | Search Cost (hours) |
|---|---|---|---|---|
| EZNAS (Akhauri et al., 2022) | 0.65 | 0.65 | 0.61 | 24.0 |
| UP-NAS (Huang et al., 2024) | 0.71 | - | - | 0.03 |
| SR-NAS (Ours) | **0.76** | **0.76** | **0.72** | 0.17 |

scores and search costs between our framework and other frameworks that also automate the design of ZC metrics, specifically EZNAS and UP-NAS. The results indicate that our metric consistently yields the highest Kendall's $\tau$ scores across all three problems in NAS-Bench-201. Notably, our framework outperforms EZNAS, even though both frameworks utilize Symbolic Regression (SR) as the search algorithm. This suggests the effectiveness of using high-level features (i.e., hand-crafted ZC metrics) over low-level features (i.e., network statistics) to design a robust ZC metric. On the other hand, the benefit of using SR in synthesizing ZC metrics is highlighted by our higher $\tau$ scores compared to UP-NAS, despite both frameworks using hand-crafted ZC metrics as input features.

In terms of search cost, our framework (0.17 hours ≈ 10 minutes) and UP-NAS (0.03 hours ≈ 2 minutes) are significantly lower than EZ-NAS (24 hours) due to leveraging the NAS benchmarks. Although our time is slightly higher than UP-NAS (i.e., 8 minutes), it is important to note that our framework only requires a single search, and our obtained ZC metric can be employed as an effective proxy across different NAS problems. In contrast, UP-NAS requires re-conducting the search process for each specific problem.

### 4.3 Capability of searching top-performing networks

Table 3: Kendall's Tau rank correlation and test accuracy on ImageNet of the best networks found by ZC metrics within Once-For-All search spaces. Best results are presented in **bold** format.

| Metric | Kendall's Tau | Top-1 Accuracy (%) |
|---|---|---|
| MeCo (Jiang et al., 2023) | 0.51 | 76.40 |
| Zen (Lin et al., 2021) | 0.59 | 76.43 |
| FLOPs | 0.60 | 76.30 |
| Snip (Lee et al., 2019) | 0.04 | 76.13 |
| L2-norm (Abdelfattah et al., 2021) | 0.41 | 76.51 |
| ZiCo (Li et al., 2023) | **0.68** | 76.44 |
| **Ours** | 0.60 | **76.65** |
| *Optimal (in 1000 networks)* | - | *76.81* |

Table 4: Comparison of error rates (%) on the CIFAR-10 dataset for networks discovered in the DARTS search space. For training-free methods, the specific ZC metrics used by the algorithms are listed in parentheses. The search cost is measured in GPU days.

| Method | Test Error (%) | Params (M) | Search Cost | Training-free |
|---|---|---|---|---|
| PNAS (Liu et al., 2018) | 3.41 ± 0.09 | 3.2 | 225 | |
| DARTS (Liu et al., 2019) | 3.00 ± 0.14 | 3.3 | 4.0 | |
| RandomNAS (Li & Talwalkar, 2020) | 2.85 ± 0.08 | 4.3 | 2.7 | |
| DARTS-PT (Wang et al., 2021) | 2.61 ± 0.08 | 3.0 | 0.8 | |
| PreNAS (Peng et al., 2023) | 2.49 ± 0.09 | 4.5 | 0.6 | |
| PINAT (Lu et al., 2023) | 2.54 ± 0.08 | 3.6 | 0.3 | |
| TE-NAS (NLR, NTK) (Chen et al., 2021) | 2.63 ± 0.06 | 3.8 | 0.03 | ✔ |
| Zero-Cost-PT (ZiCo) (Li et al., 2023) | 2.80 ± 0.03 | 5.1 | 0.04 | ✔ |
| Zero-Cost-PT (MeCo) (Jiang et al., 2023) | 2.69 ± 0.05 | 4.2 | 0.08 | ✔ |
| SWAP-NAS[†] (SWAP) (Peng et al., 2024) | 2.48 ± 0.09 | 4.3 | 0.004 | ✔ |
| Aging Evolution (SR-NAS) (Ours) | 2.66 ± 0.04 | 3.9 | 0.01 | ✔ |

[†] SWAP-NAS searches in a sub-space of DARTS, where the normal and reduction cells are similar.

In addition to the rank correlation with network performance, the ability to identify top-performing networks is a crucial aspect of ZC metrics. Previous studies have highlighted the limitations of ZC metrics in effectively identifying top-performing networks, even when they achieve high rank-correlation scores (Jiang et al., 2023; Phan & Luong, 2024). We verify this ability of our obtained ZC metric by testing it on the large-scale ImageNet dataset. Specifically, we randomly sample 1,000 networks from the Once-For-All (OFA (Cai et al., 2020)) search space and then compute the rank correlation between our ZC metric score and the networks' accuracy on the ImageNet dataset. Table 3 reveal that our Kendall rank correlation is only slightly lower than ZiCo (i.e., 0.60 compared to 0.68). However, the network with the highest score according to our ZC metric achieves the highest top-1 accuracy among all networks identified by the competing ZC metrics. This demonstrates the ability of our synthesized metric to reliably identify high-performing networks

in large-scale search spaces such as OFA. We further employ our metric as the search objective of Aging Evolution (Real et al., 2019), which is a widely-used evolutionary algorithm for NAS. The algorithm is deployed in the DARTS search space (Liu et al., 2019) to seek high-performance networks for the CIFAR-10 dataset. As shown in Table 4, the algorithm using our SR-designed ZC metric as the search objective could figure out a network with comparable performance within 15 minutes (i.e., 0.01 GPU days) (the network architecture found by our algorithm is exhibited in Appendix E). This result demonstrates both the efficiency and the effectiveness of our obtained ZC metric in discovering high-performance networks. We additionally provide a comparison of the best networks identified by our ZC metric versus other metrics across all problems in NAS-Bench-Suite-Zero in Appendix F, showing that the networks found with our metric are the best compared to those found with hand-crafted ZC metrics in 8 out of 13 total problems.

### 4.4 ABLATION STUDY

**Extensibility:** We demonstrate the extensibility of our proposed framework by additionally experimenting with *two* variants of the current input dataset (details of each dataset are presented in Appendix A.1). The results detailed in Appendix A.1 demonstrate that augmenting the input dataset with robust ZC metrics can further enhance the performance of our proposed framework. Additionally, we hypothesize that the generalizability of our framework could be improved by using a dataset that is augmented with a broader range of problems. Moreover, the metric found by the framework using the dataset covering more problems delivers better general performance across all 13 problems in NAS-Bench-Suite-Zero than the current best metric (i.e., Equation 2).

**Lack of generalizability when searching on a single problem:** We conduct additional experiments to investigate the potential overfitting of our framework when searching with the input dataset that only covers a single problem. The results detailed in Appendix A.2 reveal that the $\tau$ scores of ZC metrics found by using the datasets that only cover a single problem significantly deteriorate when evaluated on other problems. For instance, the metric found by using the dataset built on the NB301-CF10 problem achieves a score of 0.48 for NB301-CF10 but has a negative score of -0.17 for NB201-CF10. Conversely, the ZC metric found by using the dataset containing three problems achieves a more balanced performance across all problems.

**Comparison to the search procedure in EZNAS:** Appendix A.3 presents Kendall's $\tau$ for the ZC metrics designed using our proposed framework compared to the ones designed by the framework in EZNAS on three problems: NB101-CF10, NB201-CF10, and NB301-CF10. The results show that while our obtained ZC metrics are only *slightly* worse on one problem (i.e., NB301-CF10), they are *significantly* better on the other two (i.e., NB101-CF10 and NB201-CF10), highlighting the effectiveness of our approach over EZNAS. The potential issue with EZNAS, as indicated, is that it could eliminate high-quality parents and replace them with lower-quality offspring, which can lead to the offspring in the next generations being derived from inferior candidates. The search framework of EZNAS is thus less effective than ours when searching under the same budget.

## 5 CONCLUSION

In this paper, we propose a novel framework that leverages Symbolic Regression (SR) to automatically design robust zero-cost (ZC) metrics for various NAS search spaces and tasks. Our approach introduces several notable features. First, unlike previous frameworks, the input dataset of our framework comprises diverse problems instead of a single one. We further introduce a novel evaluation mechanism that assesses the quality of generated ZC metrics throughout the search process. This mechanism not only guides SR to identify expressions that correlate well with network performance across multiple problems but also prevents it from overfitting to a specific problem. Additionally, our framework is highly extensible; incorporating more powerful hand-crafted ZC metrics and diverse problems into the input dataset could enhance the effectiveness of resulting metrics across various problems. Extensive experiments on NAS-Bench-Suite-Zero demonstrate both the efficacy and efficiency of our method. Our framework could design a ZC metric with state-of-the-art Kendall's $\tau$ correlation on NAS-Bench-101/201/301 search spaces within 10 minutes, and our metric remains competitive to hand-crafted metrics on TransNAS-Bench-101-Micro/Macro search spaces. When combining our obtained metric with an evolutionary algorithm, we could figure out a network architecture with comparable performance in the DARTS search space within 15 minutes.

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

# A  FULL RESULTS IN ABLATION STUDY

## A.1  EXTENSIBILITY

Table 5: Kendall's $\tau$ rank correlations of the best ZC metrics (over 31 runs) designed by our framework with using three different input datasets: $\mathcal{D}$, $\mathcal{D}^-$, and $\mathcal{D}^+$. Best results for each problem are presented in red color.

| Dataset | NB101-CF10 | NB201-CF10 | NB301-CF10 | TNB101-Micro-Scene | TNB101-Macro-Scene |
|---------|-----------|-----------|-----------|-------------------|-------------------|
| $\mathcal{D}^-$ | 0.54 | 0.72 | 0.39 | 0.19 | -0.30 |
| $\mathcal{D}^+$ | 0.48 | **0.78** | **0.42** | **0.63** | **0.67** |
| $\mathcal{D}$ | **0.61** | 0.76 | 0.40 | 0.48 | 0.53 |

In this section, we present the full results of experiments to verify the extensibility of our proposed framework (i.e., Section 4.4). Specifically, we experiment with three following input datasets:

1. $\mathcal{D}$: The dataset used in Section 4.1 to search a robust ZC metric across multiple problems. This dataset contains the scores of 16 hand-crafted ZC metrics (including 13 ZC metrics in NAS-Bench-Suite-Zero and three newly-proposed metrics (i.e., ZiCo, MeCo, SWAP)) on **NB101-CF10**, **NB201-CF10**, and **NB301-CF10** problems.

2. $\mathcal{D}^-$: We remove the scores of three novel hand-crafted ZC metrics (i.e., ZiCo, MeCo, and SWAP) from $\mathcal{D}$.

3. $\mathcal{D}^+$: We integrate into $\mathcal{D}$ the ZC metrics scores and networks performance on two additional problems: **TNB101-Micro-Scene** and **TNB101-Macro-Scene**.

For each dataset, we execute 31 independent runs with the settings as presented in Section 4.1. Table 5 exhibits the Kendall's $\tau$ scores of the best ZC metrics designed by our framework with $\mathcal{D}$, $\mathcal{D}^-$, and $\mathcal{D}^+$ input datasets. The results demonstrate that using dataset $\mathcal{D}$ achieves higher Kendall's $\tau$ scores across all testing problems than using dataset $\mathcal{D}^-$. This suggests that augmenting the input dataset with robust ZC metrics can further enhance the performance of our proposed framework. Additionally, we hypothesize that the generalizability of our framework could be improved by using a dataset that is augmented with a broader range of problems. While the $\tau$ score for the NB101-CF10 problem slightly decreases when using dataset $\mathcal{D}^+$, the $\tau$ scores or the remaining problems show an increase compared to the framework using dataset $\mathcal{D}$. Moreover, the metric found by the framework using $\mathcal{D}^+$ delivers better general performance across all 13 problems in NAS-Bench-Suite-Zero than the current best metric (i.e., Equation 2), which is found by using $\mathcal{D}$.

## A.2  LACK OF GENERALIZABILITY WHEN SEARCHING ON A SINGLE PROBLEM

Table 6: Comparisons in Kendall's $\tau$ rank correlations of the best ZC metrics (over 31 runs) designed by our framework using the input datasets that only cover a single problem (i.e., NB101-CF10, NB201-CF10, and NB301-CF10) and using the input dataset containing all 3 problems NB101-CF10, NB201-CF10, and NB301-CF10. Best results are presented in red color.

| Search Problem | NB101-CF10 | NB201-CF10 | NB301-CF10 |
|---------------|-----------|-----------|-----------|
| NB101-CF10 | **0.66** | 0.70 | 0.37 |
| NB201-CF10 | 0.47 | **0.79** | 0.27 |
| NB301-CF10 | 0.36 | -0.17 | **0.48** |
| All 3 problems | 0.61 | 0.76 | 0.40 |

We conduct additional experiments to investigate the potential overfitting of our framework when searching with the input dataset that only covers a single problem. Specifically, we execute our framework with the input datasets that only cover a single problem (i.e., NB101-CF10, NB201-CF10, and NB301-CF10) and conduct a performance comparison to the framework that uses the

input dataset containing simultaneously three problems. Table 6 reveals that the $\tau$ scores of ZC metrics found by using the datasets that only cover a single problem significantly deteriorate when evaluated on other problems. For instance, the metric found by using the dataset built on the NB301-CF10 problem achieves a score of 0.48 for NB301-CF10 but has a negative score of -0.17 for NB201-CF10. Conversely, the ZC metric found by using the dataset containing three problems achieves a more balanced performance across all problems.

## A.3 COMPARISON TO THE SEARCH PROCEDURE IN EZNAS

Table 7: Comparisons in the Kendall's $\tau$ scores of ZC metrics designed by our framework and the framework proposed in EZNAS. Best results are presented in red color.

|        | Problem   | EZNAS      | Ours       |
|--------|-----------|------------|------------|
| Worst  | NB101-CF10 | 0.5057    | **0.5338** |
|        | NB201-CF10 | 0.7279    | **0.7363** |
|        | NB301-CF10 | **0.3684** | 0.3537    |
| Median | NB101-CF10 | 0.5372    | **0.5613** |
|        | NB201-CF10 | 0.7345    | **0.7617** |
|        | NB301-CF10 | **0.3779** | 0.3760    |
| Best   | NB101-CF10 | 0.5843    | **0.6145** |
|        | NB201-CF10 | 0.7396    | **0.7610** |
|        | NB301-CF10 | **0.4040** | 0.4012    |

Table 7 presents Kendall's $\tau$ for the ZC metrics designed using our proposed framework compared to the ones designed by the framework in EZNAS on three problems: NB101-CF10, NB201-CF10, and NB301-CF10. We compare both frameworks in three kinds of ZC metrics: the worst, the median, and the best among 31 designed metrics. The results show that while our SR-designed ZC metrics are only *slightly* worse on one problem (i.e., NB301-CF10), they are *significantly* better on the other two (i.e., NB101-CF10 and NB201-CF10), highlighting the effectiveness of our approach over EZNAS.

## B   MATHEMATICAL OPERATORS

In our experiments, we implement the primitive mathematical operators for SR: {*add*, *sub*, *mul*, *div*, *neg*, *log*, *sqrt*}. The details of each operator are presented in Table 8. Among these operators, three operators have constraints in these input values: *div* (i.e., $Y \neq 0$), *log* (i.e., $X > 0$), and *sqrt* (i.e., $X \geq 0$). We return the value of -10,000,000 for these operators in cases where the input values violate constraints.

Table 8: List of mathematical operators

| Name | Operation | Type | Description |
|------|-----------|------|-------------|
| *add* | Addition | Binary | $Z = X + Y$ |
| *sub* | Subtraction | Binary | $Z = X - Y$ |
| *mul* | Multiplication | Binary | $Z = X \times Y$ |
| *div* | Division | Binary | $Z = X \div Y$ |
| *log* | Logarithm | Unary | $Z = \log(X)$ |
| *sqrt* | Square root | Unary | $Z = \sqrt{X}$ |
| *neg* | Negative | Unary | $Z = -X$ |

## C   LIST OF PROBLEMS

Table 9: NAS problem description

| Problem | Search Space | Task |
|---------|-------------|------|
| NB101-CF10 | NAS-Bench-101 | Image classification on CIFAR-10 dataset |
| NB201-CF10 | NAS-Bench-201 | Image classification on CIFAR-10 dataset |
| NB201-CF100 | | Image classification on CIFAR-100 dataset |
| NB201-IMGNT | | Image classification on ImageNet16-120 dataset |
| NB301-CF10 | NAS-Bench-301 | Image classification on CIFAR-10 dataset |
| TNB101-Micro-Scene | TransNAS-Bench-101-Micro | Scene classification on Taskonomy dataset |
| TNB101-Micro-Object | | Object detection on Taskonomy dataset |
| TNB101-Micro-AutoEnc | | Autoencoding on Taskonomy dataset |
| TNB101-Micro-Jigsaw | | Jigsaw puzzle on Taskonomy dataset |
| TNB101-Macro-Scene | TransNAS-Bench-101-Macro | Scene classification on Taskonomy dataset |
| TNB101-Macro-Object | | Object detection on Taskonomy dataset |
| TNB101-Macro-AutoEnc | | Autoencoding on Taskonomy dataset |
| TNB101-Macro-Jigsaw | | Jigsaw puzzle on Taskonomy dataset |

# D HYPERPARAMETER SETTING

Table 10: Hyperparameters of Symbolic Regression

| Hyperparameter | Value |
| --- | --- |
| Population Size | 100 |
| Tournament Size | 2 |
| Probability of Crossover | 0.7 |
| Probability of Subtree Mutation | 0.1 |
| Probability of Hoist Mutation | 0.5 |
| Probability of Point Mutation | 0.1 |
| Maximum number of generations | 50 |
| Minimum depth of expression tree | 2 |
| Maximum depth of expression tree | 10 |

Table 11: Hyperparameters of Aging Evolution

| Hyperparameter | Value |
| --- | --- |
| Population Size | 100 |
| Probability of Mutation | 0.5 |
| Maximum number of evaluations | 3000 |

# E NETWORK ARCHITECTURE FOUND IN DARTS SEARCH SPACE

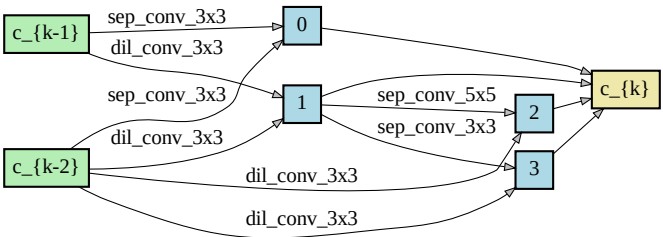

(a) Normal cell

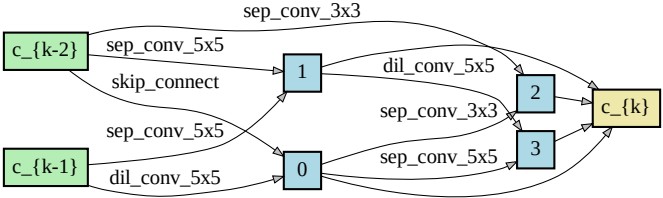

(b) Reduction cell

Figure 7: Normal and reduction cells of the architecture found by our SR-designed ZC metric in the DARTS search space for the CIFAR-10 dataset. The model size of this network is 3.97 MB.

# F PERFORMANCE COMPARISONS OF THE BEST NETWORKS FOUND BY ZC METRICS FOR PROBLEMS IN NAS-BENCH-SUITE-ZERO

Table 12: Performance comparisons of the best networks found by ZC metrics for problems within NAS-Bench-101, NAS-Bench-201, and NAS-Bench-301 search spaces. Best results for each problem are presented in red color.

| Metric | NB101-CF10 ↑ | NB201-CF10 ↑ | NB201-CF100 ↑ | NB201-IMGNT ↑ | NB301-CF10 ↑ |
|---|---|---|---|---|---|
| Params | 89.11 | 90.36 | 71.34 | 41.23 | 91.98 |
| FLOPs | 89.11 | 90.36 | 71.34 | 41.23 | 91.98 |
| Snip (Lee et al., 2019) | 86.78 | 81.45 | 47.62 | 23.40 | 91.98 |
| Fisher (Turner et al., 2020) | 86.78 | 81.45 | 47.62 | 25.93 | 91.09 |
| Jacov (Mellor et al., 2020) | 84.98 | 89.03 | 69.78 | 43.50 | 92.12 |
| Synflow (Tanaka et al., 2020) | 89.11 | 90.36 | 71.34 | 41.23 | 93.79 |
| Grasp (Wang et al., 2020)] | 86.21 | 81.45 | 47.62 | 27.60 | 92.91 |
| Plain (Abdelfattah et al., 2021) | 86.69 | 85.21 | 55.54 | 17.70 | 91.81 |
| Grad-norm (Abdelfattah et al., 2021) | 86.78 | 81.45 | 47.62 | 20.67 | 93.34 |
| EPE-NAS (Lopes et al., 2021) | 92.00 | 85.74 | 56.36 | 42.37 | 92.69 |
| L2-norm (Abdelfattah et al., 2021) | 92.56 | 88.73 | 72.04 | 45.43 | 91.98 |
| Zen (Lin et al., 2021) | 94.22 | 86.96 | 68.26 | 40.60 | 91.98 |
| NWOT (Mellor et al., 2021) | 93.33 | 89.78 | 70.14 | 45.90 | 93.54 |
| ZiCo (Li et al., 2023) | 94.22 | 90.36 | 71.34 | 41.23 | 91.98 |
| MeCo (Jiang et al., 2023) | 93.34 | 89.86 | 70.78 | 41.23 | 91.98 |
| SWAP (Peng et al., 2024) | 92.56 | 88.28 | 69.46 | 37.63 | 93.54 |
| **Ours** | **94.58** | **91.18** | **72.72** | **46.60** | **94.57** |
| *Optimal (in the benchmark)* | *94.72* | *91.57* | *73.26* | *47.33* | *94.69* |

Table 13: Performance comparisons of the best networks found by ZC metrics for problems within TransNAS-Bench-101-Micro search space. Best results for each problem are presented in red color.

| Metric | Scene ↑ | Object ↑ | Jigsaw ↑ | AutoEnc ↑ |
|---|---|---|---|---|
| Params | 53.67 | 42.14 | 85.91 | 0.46 |
| FLOPs | 53.67 | 42.14 | 85.91 | 0.46 |
| Snip (Lee et al., 2019) | 48.72 | 36.92 | 80.32 | 0.33 |
| Fisher (Turner et al., 2020) | 48.72 | 36.92 | 83.49 | 0.00 |
| Jacov (Mellor et al., 2020) | 53.72 | 41.78 | **93.99** | 0.42 |
| Synflow (Tanaka et al., 2020) | 53.67 | 39.53 | 90.91 | 0.46 |
| Grasp (Wang et al., 2020)] | 50.19 | 31.17 | 91.08 | 0.33 |
| Plain (Abdelfattah et al., 2021) | 31.51 | 31.79 | 0.03 | 0.06 |
| Grad-norm (Abdelfattah et al., 2021) | 48.72 | 36.92 | 6.82 | 0.36 |
| EPE-NAS (Lopes et al., 2021) | 52.12 | 40.86 | 92.24 | 0.46 |
| L2-norm (Abdelfattah et al., 2021) | 53.30 | 42.74 | 91.84 | 0.45 |
| Zen (Lin et al., 2021) | 53.67 | 42.14 | 85.91 | 0.46 |
| NWOT (Mellor et al., 2021) | 53.16 | **43.02** | 92.29 | 0.41 |
| ZiCo (Li et al., 2023) | 53.67 | 42.14 | 85.91 | 0.46 |
| MeCo (Jiang et al., 2023) | 52.89 | 42.16 | 91.42 | 0.46 |
| SWAP (Peng et al., 2024) | 21.95 | 40.24 | 6.82 | 0.23 |
| **Ours** | **54.24** | 42.51 | 90.62 | **0.52** |
| *Optimal (in the benchmark)* | *54.94* | *45.59* | *95.37* | *0.58* |

Table 14: Performance comparisons of the best networks found by ZC metrics for problems within TransNAS-Bench-101-Macro search space. Best results for each problem are presented in red color.

| Metric | Scene ↑ | Object ↑ | Jigsaw ↑ | AutoEnc ↑ |
|---|---|---|---|---|
| Params | 54.26 | 42.71 | 94.26 | 0.37 |
| FLOPs | 55.52 | 45.29 | **96.51** | 0.66 |
| Snip (Lee et al., 2019) | 54.50 | 42.55 | 95.20 | 0.64 |
| Fisher (Turner et al., 2020) | 51.41 | 42.55 | 93.15 | 0.44 |
| Jacov (Mellor et al., 2020) | 56.01 | 46.05 | 95.20 | 0.65 |
| Synflow (Tanaka et al., 2020) | **56.27** | 45.29 | **96.51** | 0.41 |
| Grasp (Wang et al., 2020)] | 50.36 | 41.79 | 91.55 | 0.39 |
| Plain (Abdelfattah et al., 2021) | 51.70 | 43.33 | 92.09 | 0.41 |
| Grad-norm (Abdelfattah et al., 2021) | 53.01 | 42.55 | 93.48 | 0.64 |
| EPE-NAS (Lopes et al., 2021) | 52.61 | 45.22 | 92.93 | 0.41 |
| L2-norm (Abdelfattah et al., 2021) | 56.00 | 44.50 | 94.26 | 0.58 |
| Zen (Lin et al., 2021) | 56.00 | 44.50 | 95.35 | 0.62 |
| NWOT (Mellor et al., 2021) | 55.52 | 45.29 | **96.51** | 0.66 |
| ZiCo (Li et al., 2023) | 52.43 | 42.79 | 95.35 | 0.58 |
| MeCo (Jiang et al., 2023) | **56.27** | **46.66** | **96.51** | 0.64 |
| SWAP (Peng et al., 2024) | 55.52 | 45.29 | **96.51** | 0.66 |
| **Ours** | 54.78 | 45.74 | 96.50 | **0.73** |
| *Optimal (in the benchmark)* | *57.41* | *47.42* | *97.02* | *0.75* |

# G VISUALIZATION OF RANK CORRELATION BETWEEN OUR SR-DESIGNED ZC METRIC AND THE NETWORK PERFORMANCE

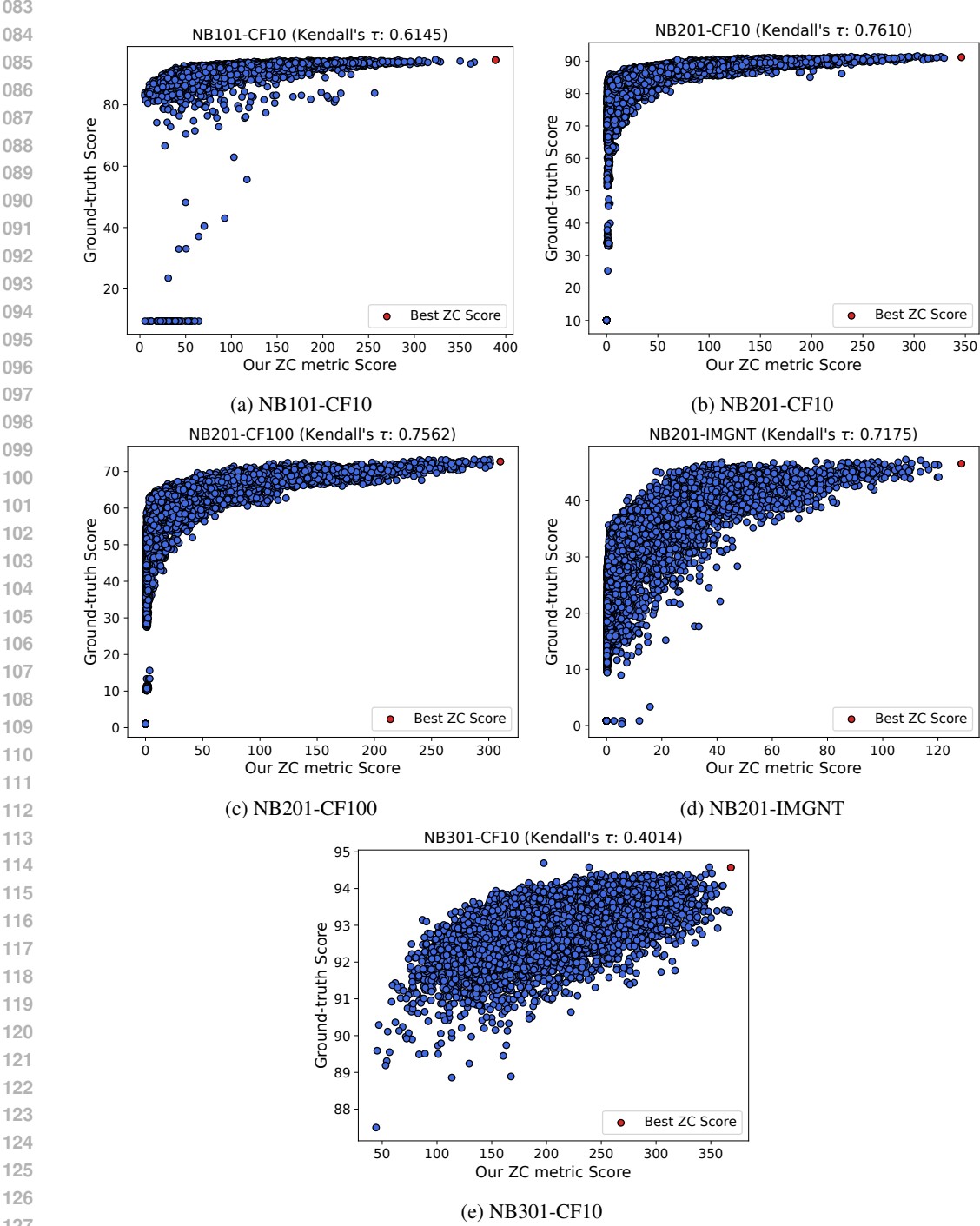

(a) NB101-CF10

(b) NB201-CF10

(c) NB201-CF100

(d) NB201-IMGNT

(e) NB301-CF10

Figure 8: Visualizations of the network performance and our ZC metric scores for the problems within the NAS-Bench-101, NAS-Bench-201, and NAS-Bench-301 search spaces.

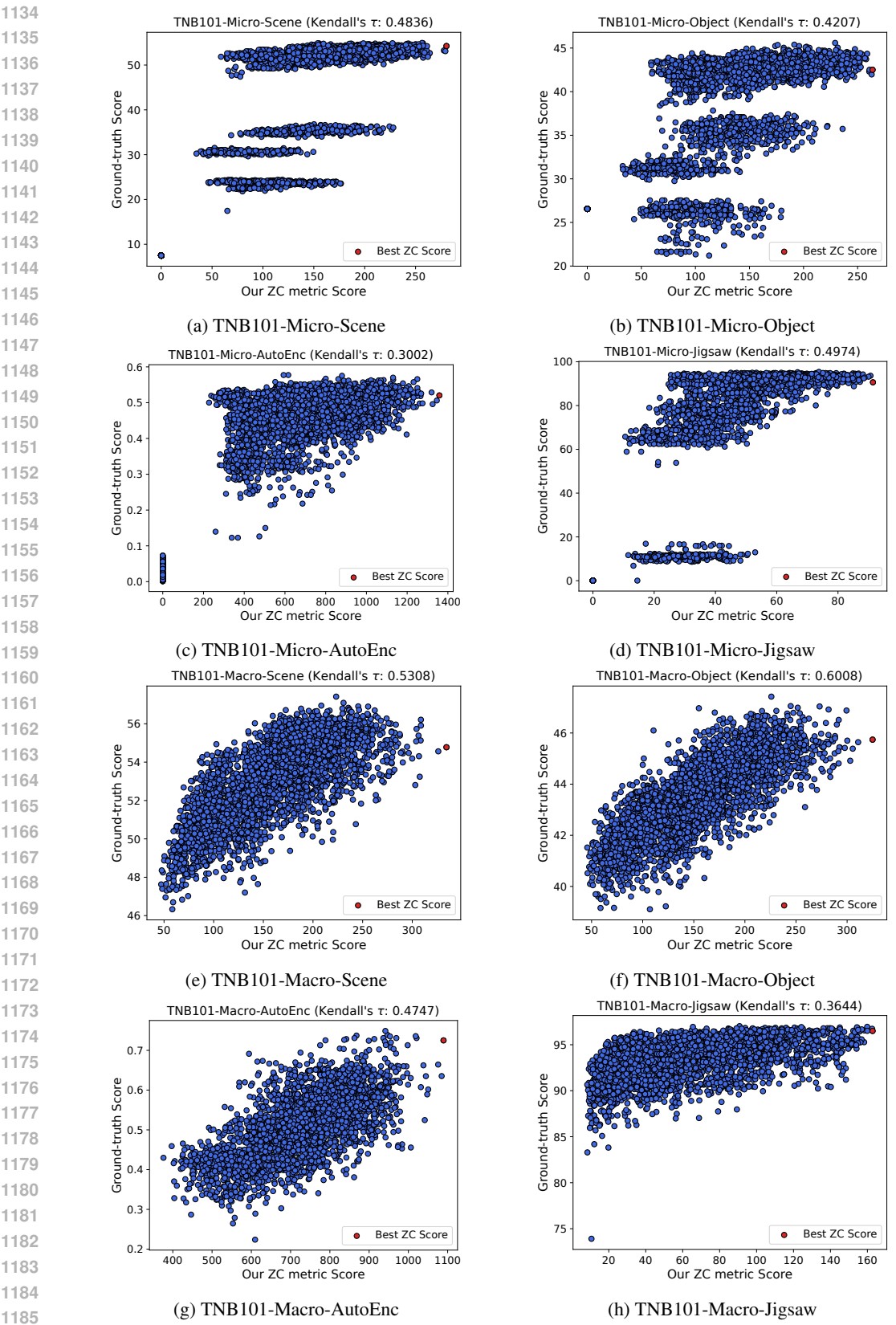

Figure 9: Visualizations of the network performance and our ZC metric scores for the problems within the TransNAS-Bench-101-Micro and TransNAS-Bench-101-Macro search spaces.

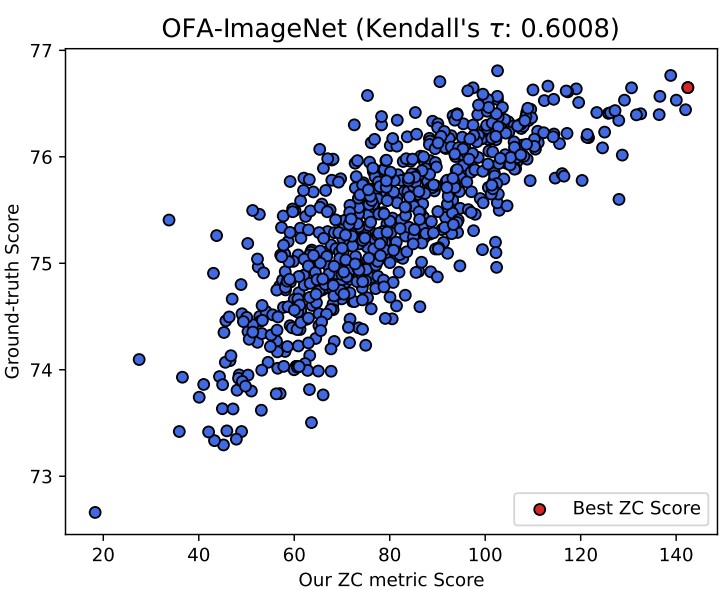

Figure 10: Visualizations of the network performance and our ZC metric scores on the ImageNet dataset for 1000 networks within the Once-For-All search space.

## H  ZERO-COST METRICS SYNTHESIZED BY OUR FRAMEWORK AT 31 RUNS

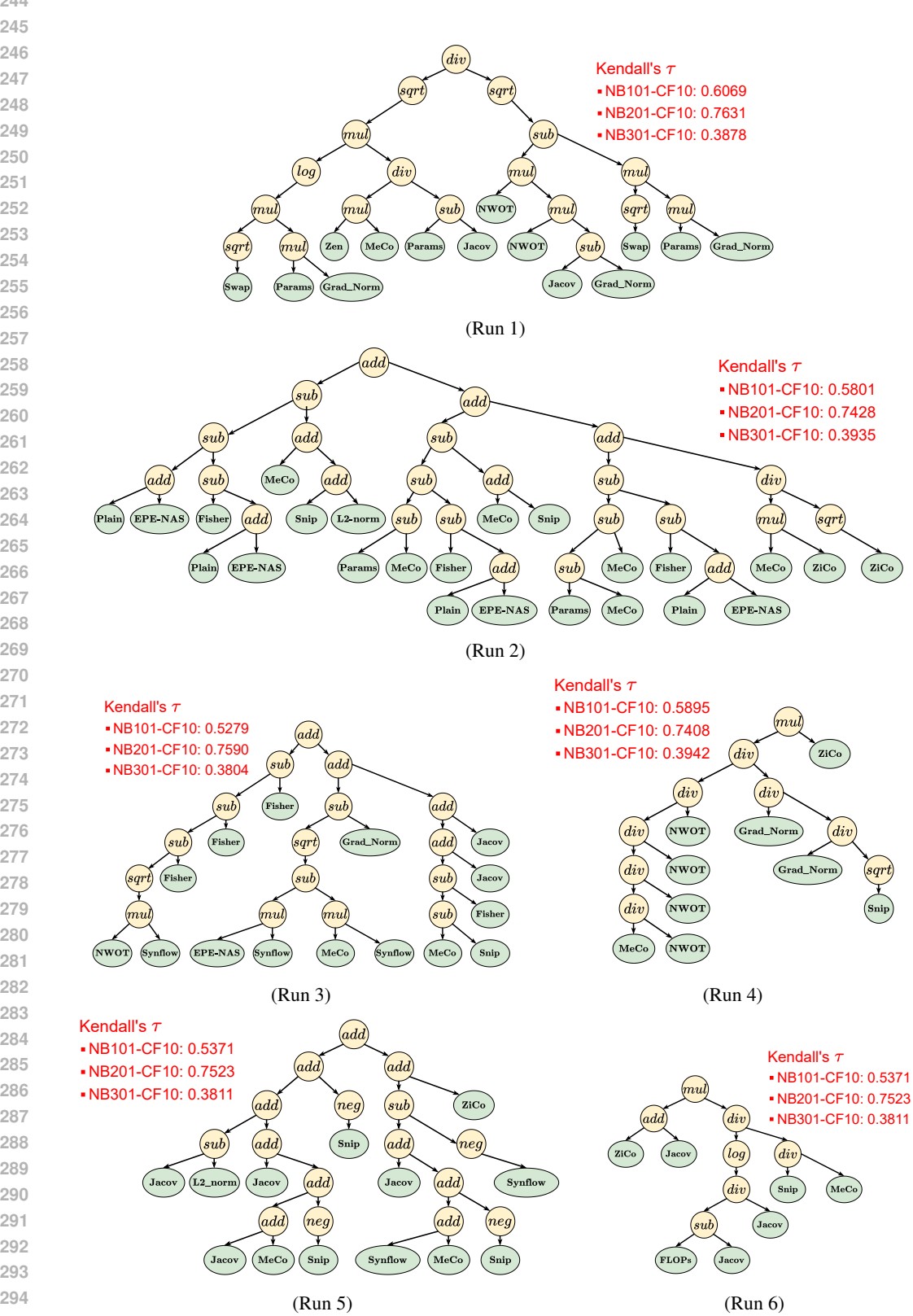

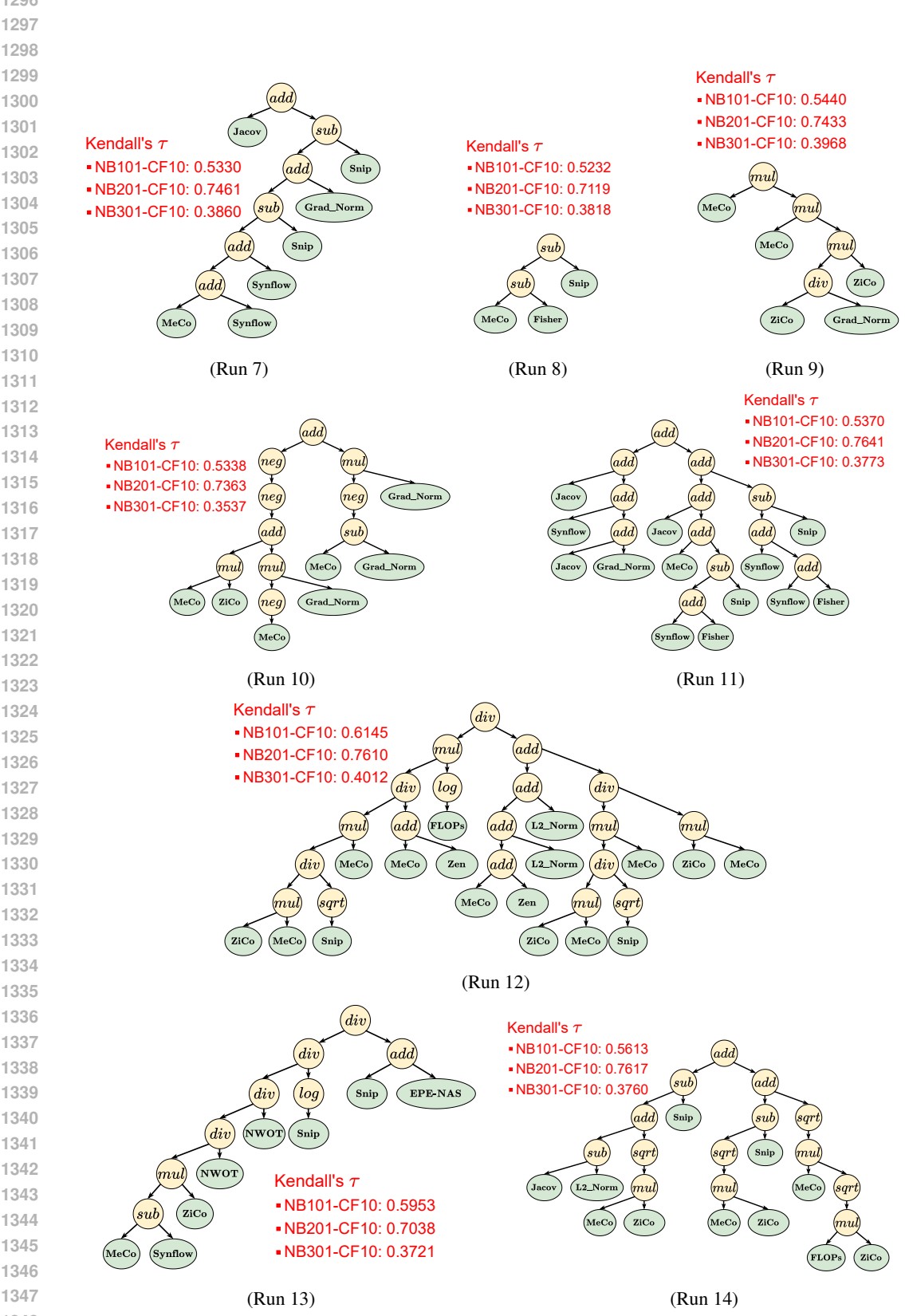

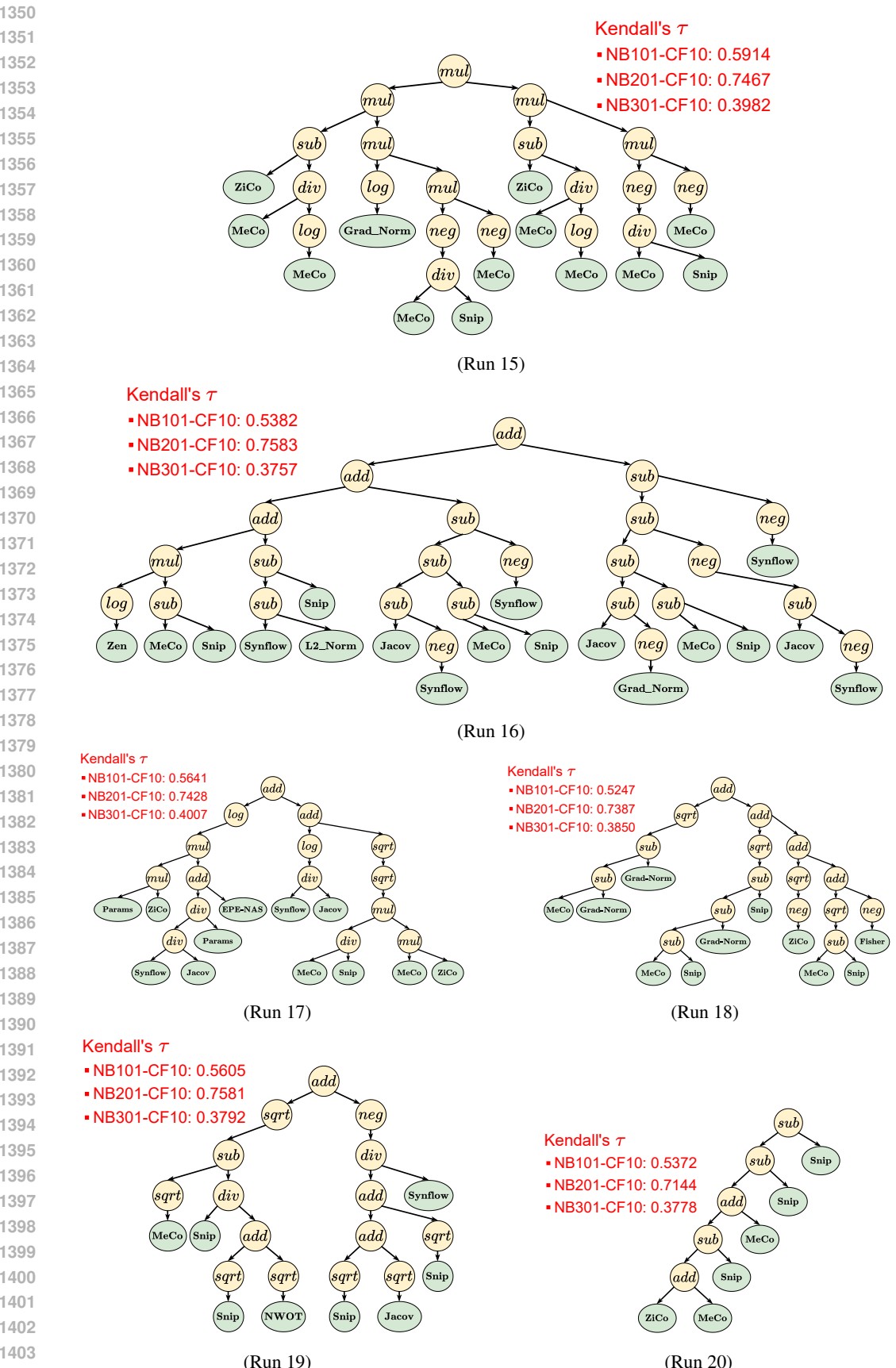

Kendall's $\tau$
- NB101-CF10: 0.5914
- NB201-CF10: 0.7467
- NB301-CF10: 0.3982

(Run 15)

Kendall's $\tau$
- NB101-CF10: 0.5382
- NB201-CF10: 0.7583
- NB301-CF10: 0.3757

(Run 16)

Kendall's $\tau$
- NB101-CF10: 0.5641
- NB201-CF10: 0.7428
- NB301-CF10: 0.4007

(Run 17)

Kendall's $\tau$
- NB101-CF10: 0.5247
- NB201-CF10: 0.7387
- NB301-CF10: 0.3850

(Run 18)

Kendall's $\tau$
- NB101-CF10: 0.5605
- NB201-CF10: 0.7581
- NB301-CF10: 0.3792

(Run 19)

Kendall's $\tau$
- NB101-CF10: 0.5372
- NB201-CF10: 0.7144
- NB301-CF10: 0.3778

(Run 20)

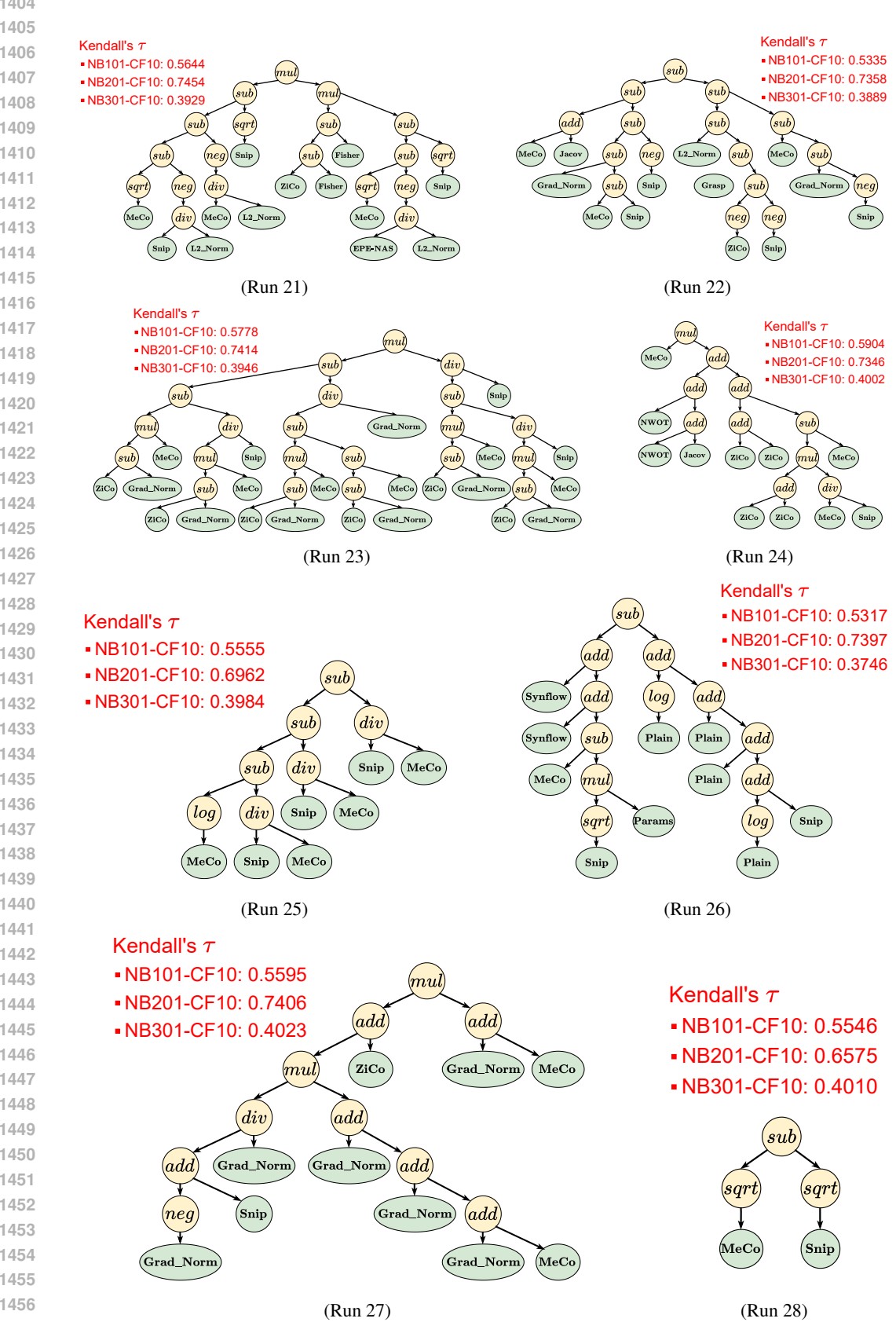

(Run 21)

(Run 22)

(Run 23)

(Run 24)

(Run 25)

(Run 26)

(Run 27)

(Run 28)

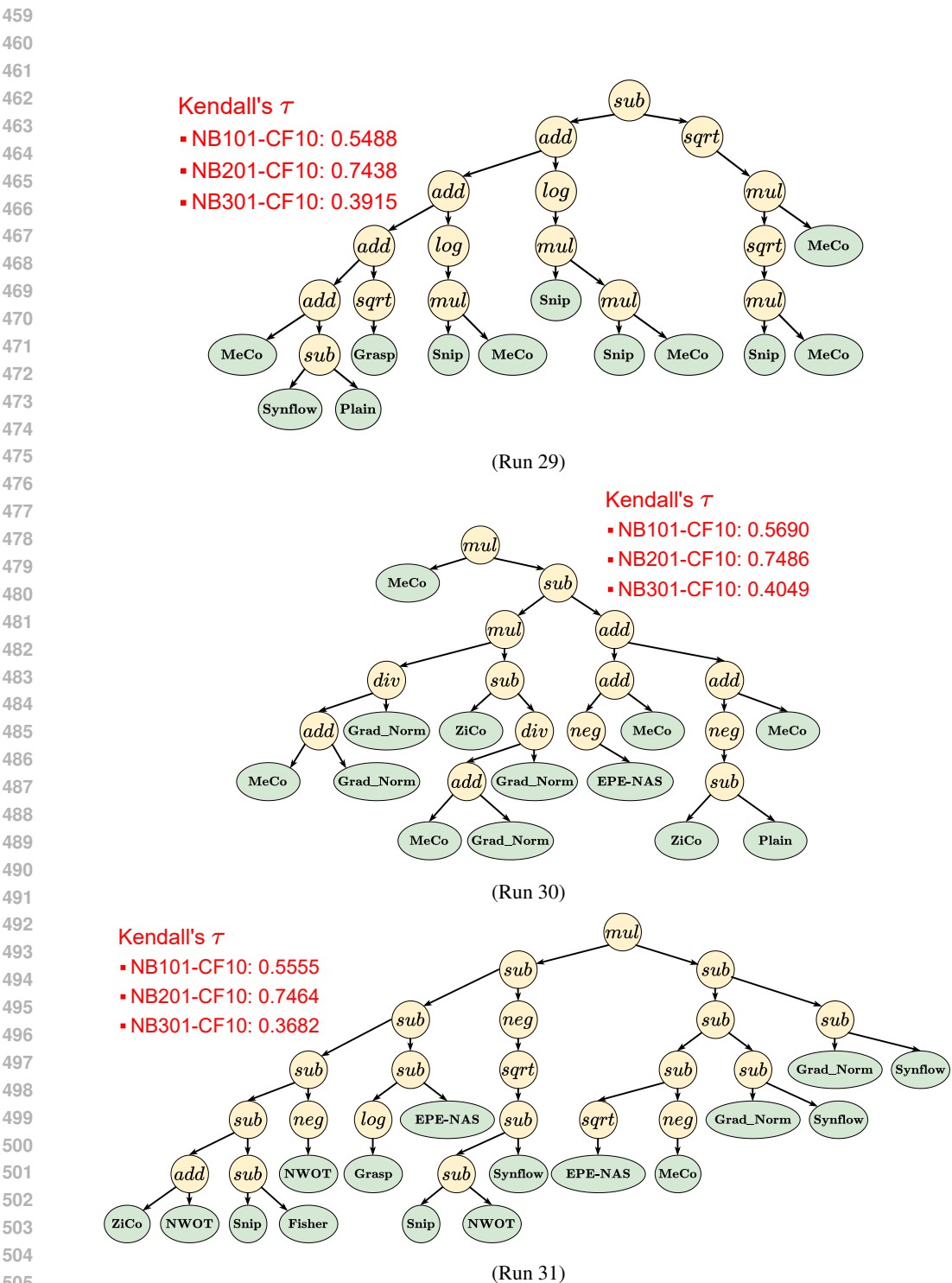

Figure 11: Expression trees of 31 ZC metrics designed by our framework. We also exhibit Kendall's $\tau$ scores across NB101-CF10, NB201-CF10, and NB301-CF10 problems for each ZC metric. The best ZC metric presented in Equation 2 is obtained at the run 12-th.

# I    FULL RESULTS IN SECTION 4.1

Table 15: Kendall's $\tau$ correlation (mean $\pm$ standard deviation) of the returned expressions over 31 runs on the input/test/(input + test) datasets for NB101/201/301-CF10 problems.

| Problem | Input | Test | Input + Test |
|---|---|---|---|
| NB101-CF10 | $0.5587 \pm 0.0258$ | $0.5603 \pm 0.0265$ | $0.5592 \pm 0.0255$ |
| NB201-CF10 | $0.7393 \pm 0.0221$ | $0.7369 \pm 0.0228$ | $0.7386 \pm 0.0222$ |
| NB301-CF10 | $0.3883 \pm 0.0133$ | $0.3864 \pm 0.0134$ | $0.3877 \pm 0.0121$ |

# J    COMPARISONS TO AZ-NAS AND RoBoT

Table 16: Comparisons to AZ-NAS and RoBoT in the Kendall's Tau rank correlation and validation accuracy (%) of the best found networks for the NB201-CF10, NB201-CF100, and NB201-IMGNET problems. Best results are presented in **red**.

| | NB201-CF10 | | NB201-CF100 | | NB201-IMGNT | |
|---|---|---|---|---|---|---|
| Metric | Kendall's Tau | Best Architecture | Kendall's Tau | Best Architecture | Kendall's Tau | Best Architecture |
| AZ-NAS (Lee & Ham, 2024) | 0.73 | 89.79 | 0.72 | 70.42 | 0.69 | 45.73 |
| RoBoT (He et al., 2024) | 0.55 | 90.93 | 0.57 | 73.18 | 0.57 | 46.43 |
| **Ours (Eqn. 2)** | **0.76** | **91.18** | **0.76** | **72.72** | **0.72** | **46.60** |
| *Optimal* (Zhao et al., 2023) | - | 91.57 | - | 73.26 | - | 47.33 |

# K    INSIGHTS FROM ANALYZING THE SYNTHESIZED ZC METRIC AND THE SEARCH PROCESS OF OUR SR FRAMEWORK

The ZC metrics used as input features for symbolic regression (SR) in this study can be categorized into two types: data-agnostic metrics (e.g., L2-Norm, Params, Synflow, and Zen) and data-dependent metrics (e.g., EPE-NAS, Fisher, FLOPs, Grad-norm, Grasp, Jacov, NWOT, Plain, SNIP, ZiCo, MeCo, and SWAP). In this section, we assess the impact of each type on the performance of our framework by comparing the best combination of data-agnostic metrics, the best combination of data-dependent metrics, and our synthesized metric (Equation 2), (which integrates both types). We note that the input dataset for SR comprises the NAS problems NB101-CF10, NB201-CF10, and NB301-CF10.

Table 17: Comparisons to the best combination of data-agnostic ZC metrics and the best combination of data-dependent ZC metrics for the NB101-CF10, NB201-CF10, NB301-CF10, and TNB101-Micro-Scene. Best and worst results are presented in **red** and **blue**, respectively.

| ZC Metric | NB101-CF10 | NB201-CF10 | NB301-CF10 | TNB101-Micro-Scene |
|---|---|---|---|---|
| Data-Agnostic | **0.41** | **0.60** | **0.31** | 0.41 |
| Data-Dependent | 0.44 | 0.71 | 0.39 | **0.20** |
| **Ours (Eqn. 2)** | **0.61** | **0.76** | **0.40** | **0.48** |

As shown in Table 17, the combination of data-dependent metrics is better than the combination of data-agnostic ones for these problems included in the input dataset (i.e., NB101-CF10, NB201-CF10, and NB301-CF10). However, when applied to TNB101-Micro-Scene (which is the problem outside the input dataset), the data-agnostic metrics demonstrate better performance. This suggests that relying solely on data-dependent metrics may lead to overfitting to the problems in the input dataset, reducing their effectiveness when applied to unseen problems. Conversely, while data-agnostic metrics show strong generalizability across diverse problems, their overall performance

is less impressive. By simultaneously incorporating both types of metrics, our synthesized metric achieves not only superior performance on problems within the input dataset but also demonstrates high generalizability and strong performance across a variety of unseen problems.

We also obtain some interesting findings when comparing the best expression tree in the first generation (Equation 3) to the best one in the final generation (Equation 2).

$$f(\cdot) = \frac{ZiCo}{L2\text{-}norm} \times \sqrt{MeCo} \qquad (3)$$

First, there is a noticeable increase in complexity, with the number of metrics rising from 3 (ZiCo, L2-norm, MeCo) to 6 (ZiCo, MeCo, Zen, SNIP, L2-norm, FLOPs). We suppose that this increased complexity enables the synthesized metric to capture more characteristics of architectures. Second, the core metrics ZiCo, L2-norm, and MeCo are consistently retained across both versions, demonstrating that the SR model effectively identifies and preserves metrics that contribute significantly to high performance. Lastly, the increased complexity results in substantial improvements in Kendall's Tau scores, indicating a stronger correlation with true performance: from 0.39, 0.70, and 0.35 (for the initial synthesized metric) to 0.61, 0.76, and 0.40 (for the final synthesized metric) across three NAS problems.

We also explore the impact of replacing the most frequently used ZC metrics in our synthesized metric (e.g., ZiCo, Snip, and MeCo in Equation 2) with the least frequently used ones (e.g., SWAP, Grasp, and Plain) and compare the Kendall's Tau scores of these two variants. As shown in Table 18, the effectiveness of the synthesized metric is significantly reduced in all replacement cases. This result, coupled with the presence of effective metrics like ZiCo and MeCo in both the initial and final populations, demonstrates that our SR framework effectively identifies these metrics as crucial components when combined with others to form potential "building blocks." The SR framework then assembles these building blocks to create high-performing ZC metrics. Therefore, substituting components within these building blocks disrupts their structures, leading to a noticeable decline in the performance of the synthesized metric.

Table 18: Kendall's Tau score of our best synthesized ZC metric (Equation 2 when replacing the most frequently used ZC metrics (i.e., ZiCo, Snip, and MeCo) with the least frequently used ZC metrics (SWAP, Grasp, and Plain). The results are colored **blue** if the replacement causes a decreasing in performance.

| Replacing | NB101-CF10 | NB201-CF10 | NB301-CF10 |
|---|---|---|---|
| MeCo → SWAP | **0.52** | **0.41** | **0.39** |
| ZiCo → Grasp | **0.15** | **0.45** | **0.22** |
| SNIP → Plain | **0.43** | **0.71** | **0.33** |
| **Ours (Eqn. 2)** | **0.61** | **0.76** | **0.40** |

