# OpenReview forum: "Crafting Zero-Cost Proxy Metrics for Neural Architecture Search via Symbolic Regression"
_ICLR.cc/2025/Conference — Submitted to ICLR 2025_

### Official Review · Reviewer_ubay · 2024-10-15

**Soundness:** 2
**Presentation:** 3
**Contribution:** 2
**Rating:** 5
**Confidence:** 4

**Summary:**

This work proposes an extensible framework for the automatic discovery of ZC metrics, where SR is used to guide the production of new metrics.

**Strengths:**

The author clearly expresses the proposed method and demonstrates the advantages of the proposed method through numerous experiments.
It makes sense that many new ZC methods (e.g., ZiCo and MeCo) are integrated into the SR model.

**Weaknesses:**

I think testing on Imagenet using the DARTS search space is necessary.
The proposed method is not advantageous compared to AZ-NAS.

**Questions:**

Regarding the automatic design of ZC agents, some works have been left out [r1-r3].
[r1] Robustifying and Boosting Training-Free Neural Architecture Search
[r2] AZ-NAS: Assembling Zero-Cost Proxies for Network Architecture Search
[r3] Dynamic ensemble of low-fidelity experts: Mitigating nas “cold-start”.

---

> ### Author Response · Authors · 2024-11-22
> **Performance on ImageNet and comparison with AZ-NAS**
>
> We sincerely appreciate the time and expertise you have devoted to reviewing our paper. Your thoughtful feedback and suggestions are immensely valuable to us. In the following, we provide a detailed, point-by-point response to address your concerns and recommendations.
>
> **Weaknesses**:
> > I think testing on Imagenet using the DARTS search space is necessary.
>
> We appreciate your suggestion and agree that validating on a large-scale dataset such as ImageNet is necessary. In the revised manuscript, we have included additional experiments on the ImageNet dataset using the Once-for-All (OFA) search space. Specifically, we randomly sample 1000 networks from the large-scale OFA search space and then compute the rank correlation between our ZC metric score and the networks’ accuracy on the ImageNet dataset.
> The new results reveal that our Kendall rank correlation is only slightly lower than ZiCo (i.e., 0.60 compared to 0.68). However, the network with the highest score according to our ZC metric achieves the highest top-1 accuracy among all networks identified by the competing ZC metrics.
>
> | Metric |      Kendall's Tau|  Top-1 Accuracy |
> |:----------|:-------------:|:------:|
> | MeCo (Jiang et al., 2023)  | 0.51 | 76.40 |
> | Zen (Lin et al., 2021) | 0.59 | 76.43 |
> | FLOPs | 0.60 | 76.30 |
> | Snip (Lee et al., 2019) | 0.04 | 76.13 |
> | L2-norm (Abdelfattah et al., 2021) | 0.41 | 76.51 |
> | ZiCo (Li et al., 2023) | **0.68** | 76.44 |
> | **Ours** | 0.60 | **76.65** |
> |_Optimal (in 1000 networks)_ | - | _76.81_|
>
> > The proposed method is not advantageous compared to AZ-NAS.
>
> Thank you for raising this point. While both our method and AZ-NAS involve combining ZC metrics, we would like to clarify the fundamental differences between the two approaches.
> * **Generalizability Across Problems**
>     + In our framework, the synthesized ZC metric is derived through a one-time search process and can be directly applied to other NAS problems without the need to re-perform the search.
>     + In contrast, AZ-NAS computes a weighted sum of four ZC metrics, where the weights are dynamically calculated during each NAS run. This requires AZ-NAS to determine a new weight vector for ZC metrics every time it is applied to a new NAS problem.
>     + Our approach is designed to discover a single ZC metric combination that demonstrates strong generalizability across diverse NAS problems, whereas AZ-NAS is more problem-specific in its application.
> * **Experimental Comparison**
> To provide a fair comparison, we have reproduced the AZ-NAS results for three NAS problems in the NAS-Bench-201 search space (NB201-CF10, NB201-CF100, and NB201-IMGNT). The results compare the Kendall’s Tau rank correlation and the performance of the best-found network architectures.
>    + Our synthesized ZC metric outperforms AZ-NAS on all three problems in both the Kendall’s Tau score and the accuracy of the best found networks.
>    + Notably, this includes NB201-CF100 and NB201-IMGNT, which were not part of the dataset used to synthesize our metric. This highlights the robustness of our approach, as opposed to AZ-NAS, where the weights are recalibrated for each specific problem.
>
> |        | NB201-CF10                     || NB201-CF100                    || NB201-IMGNT                    ||
> |--------|:-------------------:|:-------------------:|:--------------------:|:-------------------:|:--------------------:|:-------------------:|
> | **Metric** | Kendall's Tau | Best Architecture | Kendall's Tau | Best Architecture | Kendall's Tau | Best Architecture |
> | AZ-NAS (Lee & Ham, 2024) | 0.73 | 89.79 | 0.72 |  70.42 | 0.69 |  45.73 |
> | **Ours**  | **0.76** | **91.18** | **0.76** | **72.72** | **0.72** | **46.60** |
> |_Optimal (in the benchmark)_ | - | _91.57_ | - | _73.26_|  - | _47.33_|
>
> We believe these results underscore the unique strengths of our method, particularly its ability to generalize across NAS problems without problem-specific adaptations, a clear distinction from AZ-NAS. Thank you for allowing us to clarify these differences and to further strengthen the positioning of our work.

---

> ### Author Response · Authors · 2024-11-22
>
> **Questions**:
> > Regarding the automatic design of ZC agents, some works have been left out [r1-r3]. [r1] Robustifying and Boosting Training-Free Neural Architecture Search [r2] AZ-NAS: Assembling Zero-Cost Proxies for Network Architecture Search [r3] Dynamic ensemble of low-fidelity experts: Mitigating nas “cold-start”.
>
> Although our approach and [r1-r3] aim to utilize multiple ZC metrics, there are clear differences between our method and these studies. Notably, these works focus on problem-specific approaches, while our framework is designed to synthesize a generalizable ZC metric that can be directly applied across various NAS problems without re-performing the search process.
> * AZ-NAS [r2] computes a weighted sum of four ZC metrics, where the weights are dynamically optimized during each NAS run for solving a specific NAS problem. This makes AZ-NAS inherently problem-specific, as it recalibrates the weights for every new NAS task. In contrast, our synthesized ZC metric is derived once and remains applicable across diverse tasks and search spaces, eliminating the need for repeated optimization.
> * RoBoT [r1] employs a method similar to UP-NAS [r4], which has already been compared in our original manuscript. In RoBoT, the combination of ZC metrics is obtained through Bayesian Optimization to find weights for six ZC metrics, using ground-truth performance data during the weight optimization process. Consequently, RoBoT’s approach is more computationally expensive than both AZ-NAS and our method due to its reliance on ground-truth performance data.
> * DELE [r3] diverges significantly from both our approach and the other referenced methods. It employs ZC metrics to train individual predictors (low-fidelity experts), where each predictor corresponds to a specific ZC metric. The final ensemble predictor is trained using the ground-truth performance of architectures. While DELE is an innovative approach, it lies outside the scope of our work as it focuses on ensemble learning and requires significant computational resources for training, making it fundamentally different from our goal of synthesizing a lightweight, training-free ZC metric.
>
> In the revised manuscript, we will include additional experiments comparing our method to AZ-NAS and RoBoT on three NAS problems (NB201-CF10, NB201-CF100, and NB201-IMGNT). The results demonstrate that our synthesized ZC metric consistently outperforms AZ-NAS and RoBoT in both Kendall’s Tau rank correlation and the accuracy of the best-found architectures. We do not compare our approach to DELE due to its distinct methodology and higher computational cost, which make it fundamentally incomparable to our training-free and lightweight framework.
>
> |        | NB201-CF10                     || NB201-CF100         |           | NB201-IMGNT |                   |
> |--------|:-------------:|:-------------------:|:-------------:|:-------------------:|:-------------:|:-------------------:|
> | **Metric** | Kendall's Tau | Best Architecture | Kendall's Tau | Best Architecture | Kendall's Tau | Best Architecture |
> | AZ-NAS (Lee & Ham, 2024) | 0.73 | 89.79 | 0.72 |  70.42 | 0.69 |  45.73 |
> | RoBoT (He et al., 2024) | 0.55 | 90.93 | 0.57 | 73.18 | 0.57 | 46.43 |
> | **Ours**  | **0.76** | **91.18** | **0.76** | **72.72** | **0.72** | **46.60** |
> |_Optimal (in the benchmark)_ | - | _91.57_ | - | _73.26_|  - | _47.33_|
>
> Thank you again for pointing out these relevant works, which allowed us to better contextualize our contributions and highlight the unique aspects of our approach.
>
> ------
> [r1] Zhenfeng He, Yao Shu, Zhongxiang Dai, and Bryan Kian Hsiang Low. Robustifying and Boosting Training-Free Neural Architecture Search. In The International Conference on Learning Representations ICLR 2024. OpenReview.net, 2024.
>
> [r2] Junghyup Lee and Bumsub Ham. AZ-NAS: Assembling Zero-Cost Proxies for Network Architecture Search. In IEEE Conference on Computer Vision and Pattern Recognition (CVPR), 2024.
>
> [r3] Junbo Zhao, Xuefei Ning, Enshu Liu, Binxin Ru, Zixuan Zhou, Tianchen Zhao, Chen Chen, Jiajin Zhang, Qingmin Liao, and Yu Wang. Dynamic Ensemble of Low-Fidelity Experts: Mitigating NAS ''Cold-Start”. In The Conference on Artificial Intelligence (AAAI), 2023.
>
> [r4] Yi-Cheng Huang, Wei-Hua Li, Chih-Han Tsou, Jun-Cheng Chen, and Chu-Song Chen. UP-NAS: Unified Proxy for Neural Architecture Search. In IEEE Conference on Computer Vision and Pattern Recognition (CVPR) Workshops, 2024.

---

### Official Review · Reviewer_c1zS · 2024-11-01

**Soundness:** 2
**Presentation:** 3
**Contribution:** 2
**Rating:** 3
**Confidence:** 4

**Summary:**

This paper proposes a higher-level zero-cost metric for neural architecture search based on symbolic regression of hand-designed zero-cost metrics. The proposed framework is extensible, consistent, and achieves a high positive rank correlation across multiple problems. Results are reported on NAS-Bench-101/201/301 and TransNAS-Bench-101-Micro/Macro as well as NAS on CIFAR-10 demonstrating competitive rank correlation.

**Strengths:**

+ The use of rank correlation across multiple ZC metrics and problems is interesting and addresses some of the common criticism of ZC metrics.

**Weaknesses:**

- Limited Novelty. Most contributions, e.g., the use of symbolic regression or the use of high-level ZC metrics, were proposed in earlier work. Their combination is somewhat novel but in light of the following points may not be sufficient.
- The claim "Our framework can synthesize a new ZC metric within only 10 minutes" is incorrect because the framework requires the calculation of the high-level ZC metrics which serve as input features, and therefore, the actual total time is the time required by each high-level ZC metric plus 10 mins. The same comment applies to the results in Table 4.
- The proposed ZC metric, while using existing ZC metrics, doesn't outright outperform these metrics (see Fig. 6), and in NAS, is outperformed by a number of existing NAS methods based on ZC metrics (see Table 4)
- No ImageNet results reported.

**Questions:**

- Why was symbolic regression chosen?
- Why was the proposed ZC metric (SR-NAS) not evaluated on ImageNet?
- If the proposed ZC on average ranks better than existing ZC metrics, why does it not outperform these metrics in NAS?

---

> ### Author Response · Authors · 2024-11-22
>
> **Weaknesses**:
> > The proposed ZC metric, while using existing ZC metrics, doesn't outright outperform these metrics (see Fig. 6), and in NAS, is outperformed by a number of existing NAS methods based on ZC metrics (see Table 4)
>
> Thank you for highlighting these points, as they have helped us better articulate the contributions and limitations of our work.
>
> * **For the Kendall Tau comparison in Fig. 6**
>   + We note that our synthesized ZC metric achieves the highest Kendall Tau rank correlation for 8 out of 13 problems and ranks among the top 5 metrics for the remaining problems. When applied to NAS, the architectures identified using our ZC metric achieve the highest performance on 8 out of 13 problems and exhibit competitive performance on the remaining ones (Tables 12-14, Appendix F).
>   + We will add the following new results for the ImageNet dataset in the revised manuscript. The new results show that our ZC metric achieves the second-best Kendall Tau rank correlation and our network found by using our synthesized ZC metric has the highest top-1 accuracy among the competitive ones. As noted in previous studies and in our paper, finding a ZC metric that is the best for all NAS problems is infeasible (No Free Lunch theorem) and we thus highlight the impressive results of our ZC metric (which is automatically synthesized by our framework) compared to state-of-the-art handcrafted ZC metrics.
>
> | Metric |      Kendall's Tau|  Top-1 Accuracy |
> |:----------|:-------------:|:------:|
> | MeCo (Jiang et al., 2023)  | 0.51 | 76.40 |
> | Zen (Lin et al., 2021) | 0.59 | 76.43 |
> | FLOPs | 0.60 | 76.30 |
> | Snip (Lee et al., 2019) | 0.04 | 76.13 |
> | L2-norm (Abdelfattah et al., 2021) | 0.41 | 76.51 |
> | ZiCo (Li et al., 2023) | **0.68** | 76.44 |
> | **Ours** | 0.60 | **76.65** |
> |_Optimal (in 1000 networks)_ | - | _76.81_|
>
> * **For the comparison results in Table 4**
>   + The algorithm using our ZC metric as the search objective could figure out the network with the better performance compared to the ones of several training-based NAS baselines such as DARTS and RandomNAS.
>   + When comparing our algorithm to training-free methods, our found network is better than the ones found by ZiCo and MeCo. The results show that our network is worse than SWAP-NAS (using SWAP metric) and TE-NAS (using NTK and NLR metrics). However, it is important to note SWAP-NAS operates in different search spaces, making direct comparisons less straightforward. Besides, the NTK and NLR metrics are not included in our dataset for training the SR because we aim to utilize the availability of published databases (NAS-Bench-Suite-Zero). Adding more diverse ZC metrics to the database and synthesizing more novelty ZC metrics are in progress.
>
> Once again, we sincerely appreciate the reviewer’s observation, which has helped us identify valuable directions for future work.

---

> ### Author Response · Authors · 2024-11-23
> **Highlight the contribution of our approach**
>
> **Weaknesses**
> > Limited Novelty. Most contributions, e.g., the use of symbolic regression or the use of high-level ZC metrics, were proposed in earlier work. Their combination is somewhat novel but in light of the following points may not be sufficient.
>
> We sincerely appreciate your thoughtful review and the opportunity to address the novelty and contributions of our work. While we acknowledge that our study shares certain similarities with prior works like EZ-NAS [1] and AutoProx [2], we believe our approach introduces significant advancements and distinct differences:
>
> - **Input Features**:
> Our framework employs high-level ZC metrics as input features, offering greater extensibility compared to the low-level features used in EZ-NAS [1] and AutoProx [2]. High-level ZC metrics are inherently more generalizable and transferable across diverse tasks, making them particularly effective for evaluating and ranking architectures in a wide range of NAS problems.
>
> - **Objective Function**:
> Unlike EZ-NAS [1], which evaluates metrics on a single NAS problem at a time, our approach considers multiple NAS problems simultaneously. This prevents overfitting to specific problems and ensures better generalizability. Moreover, our framework does not require predefined weights for each problem, unlike AutoProx [2], where such weights must be manually specified. This eliminates subjectivity and enhances the adaptability and robustness of our approach to diverse problem settings.
>
> - **Efficiency and Effectiveness**:
> In our experiments, we demonstrate that our approach outperforms EZ-NAS in both efficiency and effectiveness.
>
>     +  **Efficiency**: Our framework achieves a significantly lower search cost (~10 minutes) compared to the 24 hours reported for EZ-NAS.
>     + **Effectiveness**: The synthesized ZC metric from our approach consistently outperforms EZ-NAS across all tested problems in terms of Kendall’s Tau rank correlation. This result highlights the robustness and generalizability of our metric across diverse NAS problems.
>
> These advancements, particularly the integration of high-level ZC metrics, a novelty objective function, and superior empirical performance, position our work as a meaningful contribution to the automatic design of ZC metrics.
>
> We hope this explanation addresses your concerns and clarifies the novelty of our contributions regarding prior works.
>
> -----------
> [1] Yash Akhauri, Juan Pablo Munoz, Nilesh Jain, and Ravi Iyer. EZNAS: Evolving Zero-Cost Proxies For Neural Architecture Scoring. In Advances in Neural Information Processing Systems (NeurIPS), 2022.
>
> [2] Zimian Wei, Peijie Dong, Zheng Hui, Anggeng Li, Lujun Li, Menglong Lu, Hengyue Pan, and Dongsheng Li. Auto-Prox: Training-Free Vision Transformer Architecture Search via Automatic Proxy Discovery. In The Conference on Artificial Intelligence (AAAI), 2024.

---

> ### Author Response · Authors · 2024-11-23
> **Clarify the search cost of our approach**
>
> **Weaknesses**
> > The claim "Our framework can synthesize a new ZC metric within only 10 minutes" is incorrect because the framework requires the calculation of the high-level ZC metrics which serve as input features, and therefore, the actual total time is the time required by each high-level ZC metric plus 10 mins. The same comment applies to the results in Table 4.
>
> Thank you for raising this concern. We believe there has been some misunderstanding, and we would like to clarify the details below:
> - **Regarding the claim "Our framework can synthesize a new ZC metric within only 10 minutes"**:
>
>   + We want to emphasize that the 10 minutes refers specifically to the time required by the symbolic regression (SR) process to execute a single run. It does not include the time required to compute the high-level ZC metrics, which are used as input features for the SR process. Since we utilize the ZC scores reported in the NAS-Bench-Suite-Zero and do not recompute these scores during the search process, the total time would indeed be higher than 10 minutes if we were to recompute the ZC scores. We hope this clarification resolves your concern regarding the 10-minute claim for our framework.
>
> - **Regarding Table 4 and the reported 15-minute search cost**:
>    + The 15 minutes reported in Table 4 refers specifically to the time required to compute the ZC metric values, not the time required for the SR process to synthesize a new ZC metric. In our experiments, we utilize the previously synthesized ZC metric (from Eq. 2) to conduct the NAS search, and therefore, the time for SR synthesis is not included in the 15-minute estimate.
>
>      For clarity, the 15 minutes represents the total time to compute the ZC metrics for a set of architectures on the DARTS search space. To put this in context, AZ-NAS [1] computes 4 ZC metrics for each architecture in approximately 42 ms on the NAS-Bench-201 search space (totaling around 2 minutes for 3000 architectures), or SWAP-NAS [2] completes an NAS run in about 6 minutes on the DARTS search space. We believe this makes the reported 15 minutes for computing the ZC metrics clear, as it is directly comparable to existing methods.
>
> We hope this explanation resolves the misunderstanding regarding the time estimates in our framework. Thank you again for your insightful feedback.
>
> -----------
> [1] Junghyup Lee and Bumsub Ham. AZ-NAS: Assembling Zero-Cost Proxies for Network Architecture Search. In IEEE Conference on Computer Vision and Pattern Recognition (CVPR), 2024.
>
> [2] Yameng Peng, Andy Song, Haytham M. Fayek, Vic Ciesielski, and Xiaojun Chang. SWAP-NAS: Sample-Wise Activation Patterns for Ultra-fast NAS. In International Conference on Learning Representations (ICLR), 2024.

---

> ### Author Response · Authors · 2024-11-23
> **Results on ImageNet**
>
> **Weaknesses**
> > No ImageNet results reported.
>
> We thank the reviewer for highlighting this limitation.
> We will include additional experiments on the **ImageNet** dataset using the **Once-for-All (OFA)** [1] search space in the revised manuscript.
> Specifically, we randomly sample 1000 networks from the large-scale OFA search space and then compute the rank correlation between our ZC metric score and the networks’ accuracy on the ImageNet dataset.
>
> The new following results reveal that our Kendall rank correlation is only slightly lower than ZiCo (i.e., 0.60 compared to 0.68). However, the network with the highest score according to our ZC metric achieves the highest top-1 accuracy among all networks identified by the competing ZC metrics. This demonstrates the ability of our synthesized metric to reliably identify high-performing networks in large-scale search spaces such as OFA.
> | Metric |      Kendall's Tau|  Top-1 Accuracy |
> |:----------|:-------------:|:------:|
> | MeCo (Jiang et al., 2023)  | 0.51 | 76.40 |
> | Zen (Lin et al., 2021) | 0.59 | 76.43 |
> | FLOPs | 0.60 | 76.30 |
> | Snip (Lee et al., 2019) | 0.04 | 76.13 |
> | L2-norm (Abdelfattah et al., 2021) | 0.41 | 76.51 |
> | ZiCo (Li et al., 2023) | **0.68** | 76.44 |
> | **Ours** | 0.60 | **76.65** |
> |_Optimal (in 1000 networks)_ | - | _76.81_|
>
> -----------
>
> [1] Han Cai, Chuang Gan, Tianzhe Wang, Zhekai Zhang, and Song Han. Once-for-All: Train One Network and Specialize it for Efficient Deployment. In International Conference on Learning Representations (ICLR), 2020.

---

> ### Author Response · Authors · 2024-11-23
> **The reason of choosing Symbolic Regression**
>
> **Questions**
> > Why was symbolic regression chosen?
>
> Thank you for raising point. We selected SR for some specific reasons that align with the goals of our study. These reasons are presented as follows:
>
> 1. **Suitability for Tabular Data**:
> The NAS-Bench-Suite-Zero database, which forms the basis of our study, consists of tabular data. In exploring methods to effectively utilize this rich and diverse dataset, we found that traditional machine learning methods, such as SR, are well-suited for such data. Deep learning (DL)-based approaches, while powerful in other contexts, may require more extensive data and resources and might not leverage the inherent structure of tabular data as effectively. Empirically, our approach demonstrates superior performance compared to UP-NAS [1], which employs a DL model to optimize weights between ZC metrics. Additional experiments conducted in the revised manuscript further support the effectiveness of our method compared to other approaches that combine multiple ZC metrics.
>
> 2. **Interpretability**:
> One significant advantage of SR is its ability to produce interpretable mathematical expressions. This human-readable representation allows us to derive valuable insights from the synthesized ZC metric. For instance, in a related work (AutoProx [2]), SR-based design of ZC metrics revealed that specific weight parameters in linear layers of MLP modules significantly influence performance.
>
> We hope this response clarifies the reason of choosing SR in our paper.
>
> ----------
> [1] Yi-Cheng Huang, Wei-Hua Li, Chih-Han Tsou, Jun-Cheng Chen, and Chu-Song Chen. UP-NAS: Unified Proxy for Neural Architecture Search. In IEEE Conference on Computer Vision and Pattern Recognition (CVPR) Workshops, 2024.
>
> [2] Zimian Wei, Peijie Dong, Zheng Hui, Anggeng Li, Lujun Li, Menglong Lu, Hengyue Pan, and Dongsheng Li. Auto-Prox: Training-Free Vision Transformer Architecture Search via Automatic Proxy Discovery. In The Conference on Artificial Intelligence (AAAI), 2024.

---

> ### Author Response · Authors · 2024-11-23
>
> **Questions**
>
> > Why was the proposed ZC metric (SR-NAS) not evaluated on ImageNet?
>
> Thank you for raising this important point. In the response to Weakness #4, we have added the results for our metric on the **ImageNet** dataset within the **Once-For-All** [1] search space. We will include these new results in the revised manuscript.
>
> | Metric |      Kendall's Tau|  Top-1 Accuracy |
> |:----------|:-------------:|:------:|
> | MeCo (Jiang et al., 2023)  | 0.51 | 76.40 |
> | Zen (Lin et al., 2021) | 0.59 | 76.43 |
> | FLOPs | 0.60 | 76.30 |
> | Snip (Lee et al., 2019) | 0.04 | 76.13 |
> | L2-norm (Abdelfattah et al., 2021) | 0.41 | 76.51 |
> | ZiCo (Li et al., 2023) | **0.68** | 76.44 |
> | **Ours** | 0.60 | **76.65** |
> |_Optimal (in 1000 networks)_ | - | _76.81_|
>
> -----------
>
> [1] Han Cai, Chuang Gan, Tianzhe Wang, Zhekai Zhang, and Song Han. Once-for-All: Train One Network and Specialize it for Efficient Deployment. In International Conference on Learning Representations (ICLR), 2020.

---

> ### Author Response · Authors · 2024-11-23
>
> **Questions**
>
> > If the proposed ZC on average ranks better than existing ZC metrics, why does it not outperform these metrics in NAS?
>
> Thank you for raising your concern.
>
> - In Figure 6, we demonstrate that our synthesized ZC metric achieves an average rank better than previous competitive handcrafted ZC metrics. This highlights the strong generalizability of our metric, as it consistently performs well across diverse NAS problems spanning different search spaces and tasks.
>
>   When employing our metric in an NAS run, Table 4 shows that our metric is worse than SWAP, NTK and NLR. However, we acknowledge that the NAS performance is influenced not only by the ZC metric but also by additional factors such as the search algorithm, search space, and experimental hardware. The search strategy we employ in NAS experiments is Aging Evolution, which is similar to the strategy used in SWAP-NAS. However, the search space of SWAP-NAS is a subspace of DARTS, which is smaller and inherently different from the one we use. While TE-NAS achieves better results than our method, it employs a pruning-based search strategy, which is fundamentally different from the Aging Evolution strategy we utilize.
>
>   Similarly, while our metric outperforms ZiCo and MeCo in Table 4, the comparison is complicated by differences in search strategies (i.e., our method uses Aging Evolution, whereas theirs rely on DARTS-PT). We emphasize that the results in Table 4 demonstrate the potential of applying our synthesized metric in an NAS run when we could efficiently figure out a network with comparable performance to other ZC metrics or training-based ones.
>
> - To provide a fairer comparison of the capability of ZC metrics in identifying top-performing networks, we suggest referring to the results in Appendix F of the manuscript. Across the 13 NAS problems presented in the original manuscript, the networks with the highest ZC scores based on our metric achieve the best performance in 8 problems (see Tables 12-14). Additionally, in the newly experiments on the ImageNet dataset, the network identified by our metric achieves the highest top-1 accuracy among those found by competitive metrics. As noted in previous studies and in our paper, finding a ZC metric that is the best for all NAS problems is infeasible (No Free Lunch). Despite this, our method achieves the best results in 9 out of the 14 experimental problems (including the new ImageNet results), demonstrating superior performance compared to previous handcrafted ZC metrics. These results validate the effectiveness of our metric when applied in NAS.
>
> We hope this clarifies our contributions and contextualizes the results presented in the manuscript. Thank you again for the opportunity to address this concern.

---

### Official Review · Reviewer_yYgB · 2024-11-04

**Soundness:** 3
**Presentation:** 3
**Contribution:** 2
**Rating:** 5
**Confidence:** 4

**Summary:**

The authors proposed a framework based on Symbolic Regression to automate the design of ZC metrics, whch is not only highly extensible but also capable of quickly producing a ZC metric with a strong positive rank correlation to network performance across multiple problems within just a few minutes.

**Strengths:**

1.The approach of using symbolic regression to combine hand-crafted metrics into a superior proxy metric is innovative, setting it apart from previous methods, such as those requiring manual crafting of each metric.

2.The study have comprehensive experiments on multiple NAS benchmarks and comparisons with state-of-the-art ZC metrics.

3. The paper is generally well-structured and clear, with a logical flow from problem statement to methodology, experiments, and conclusions.

4. The proposed approach can evaluate network architectures quickly and accurately without intensive computational resources.

**Weaknesses:**

1. The innovation of the paper is limited，the framework is a common symbolic regression to search the best mathematical expressions of ZC metrics.

2. The paper focuses on the performance of the designed ZC metrics but does not provide insights into why certain combinations of metrics work better than others.

3. While the paper claims generalizability across various NAS problems, the evaluation is primarily based on a limited set of benchmarks.

**Questions:**

1. the Eq.1 is essentially a normalization, It also looks like giving each question the same weight.

2. Why specific metrics (e.g., FLOPs, Snip, ZiCo, etc.) that frequently appear in high-performing combinations? Are there theoretical or empirical reasons?

3. What is the multiple problems in dataset? Are they different tasks?

4. There seems to be a typo in the annotation of Figure 2. NWOT should be MeCo.

5. In Figure 4, what do SS^i and T^j in the data set represent?

---

> ### Author Response · Authors · 2024-11-22
> **Generalizability of the synthesized ZC metric**
>
> **Weaknesses**:
> > 3. While the paper claims generalizability across various NAS problems, the evaluation is primarily based on a limited set of benchmarks.
>
> We appreciate this feedback and agree that demonstrating broader generalizability would strengthen our claims. In the paper, we claim the high generalizability across various NAS problems of our metric due to the following points.
> * The dataset used to synthesize our metric using Symbolic Regression (SR) contains only three NAS problems: NB101-CF10, NB201-CF10, and NB301-CF10. These cover three search spaces (NAS-Bench-101, NAS-Bench-201, NAS-Bench-301) and a single task (image classification on the CIFAR-10 dataset). However, we tested our synthesized metric on 10 NAS problems that are distinct from the ones included in the SR dataset. For example, the NB201-CF100 problem shares the same search space as NB201-CF10 (i.e., NAS-Bench-201) but has a different task (image classification on CIFAR-100 dataset), or a series of problems built on the TransNAS-Bench-101 Micro/Macro search spaces differ from NB101-CF10, NB201-CF10, and NB301-CF10 in both search space (TransNAS-Bench-101 Micro/Macro) and task (e.g., Scene classification, Object detection, Autoencoding, Jigsaw puzzle).
> * Results in Figure 6 demonstrate that the Kendall Tau rank correlation of our metric is the best on 5 out of 10 problems excluded from the SR dataset. Additionally, results in Appendix F (Tables 12-14) reveal that the best networks found using our metric have the best performance on 5 out of these 10 problems.
> * Notably, our metric achieves comparable results to the best metric on the remaining problems where it is not the best.
>
> To further strengthen our claims about generalizability, we have obtained additional results of our metric on the ImageNet dataset using the Once-for-All (OFA) search space, which introduces a significantly different task and search space. Among 1000 networks sampled from the OFA search space, our metric exhibits a high Kendall rank correlation with their accuracy (0.60) which is only slightly lower than ZiCo (0.68). Moreover, the network with the highest score according to our ZC metric achieves the highest top-1 accuracy compared to those found by other ZC metrics. These new results further support the generalizability of our framework across diverse NAS problems.
>
> | Metric |      Kendall's Tau|  Top-1 Accuracy |
> |:----------|:-------------:|:------:|
> | MeCo (Jiang et al., 2023)  | 0.51 | 76.40 |
> | Zen (Lin et al., 2021) | 0.59 | 76.43 |
> | FLOPs | 0.60 | 76.30 |
> | Snip (Lee et al., 2019) | 0.04 | 76.13 |
> | L2-norm (Abdelfattah et al., 2021) | 0.41 | 76.51 |
> | ZiCo (Li et al., 2023) | **0.68** | 76.44 |
> | **Ours** | 0.60 | **76.65** |
> |_Optimal (in 1000 networks)_ | - | _76.81_|

---

> > ### Author Response · Authors · 2024-11-24
> > **Insights from the analyses of the synthesized ZC metrics and the search process of our SR framework**
> >
> > **Weaknesses**
> > > 2. The paper focuses on the performance of the designed ZC metrics but does not provide insights into why certain combinations of metrics work better than others.
> >
> > Thank you for raising this insightful concern.
> >
> > We have conducted an analysis of the resulting synthesized ZC metrics and the search process of our proposed SR framework.
> > We will attach the insights from the analyses to the revised manuscript.
> >
> > We first assess the impact of two kinds of ZC metrics (i.e., data-agnostic and data-dependent) on the performance of our framework by comparing the best combination of data-agnostic metrics, the best combination of data-dependent metrics, and our synthesized metric (Equation 2) (which integrates both types). The new following results show the combination of data-dependent metrics is better than the combination of data-agnostic ones for these problems included in the input dataset (i.e., NB101-CF10, NB201-CF10, and NB301-CF10). However, when applied to TNB101-Micro-Scene (which is the problem outside the input dataset), the data-agnostic metrics demonstrate better performance. This suggests that relying solely on data-dependent metrics may lead to overfitting to the problems in the input dataset, reducing their effectiveness when applied to unseen problems. Conversely, while data-agnostic metrics show strong generalizability across diverse problems, their overall performance is less impressive. By simultaneously incorporating both types of metrics, our synthesized metric achieves not only superior performance on problems within the input dataset but also demonstrates high generalizability and strong performance across a variety of unseen problems.
> > | ZC Metric |      NB101-CF10 |  NB201-CF10 |  NB301-CF10 |  TNB101-Micro-Scene |
> > |:----------|:-------------:|:------:|:------:|:------:|
> > | Data-Agnostic | 0.41 | 0.60 | 0.31 | 0.41 |
> > | Data-Dependent | 0.44 |  0.71 | 0.39 | 0.20 |
> > | **Ours (Equation 2)** | **0.61** | **0.76** | **0.40** | **0.48** |
> >
> > We also obtain some interesting findings when comparing the best expression tree in the first generation (i.e., **ZiCo * L2-norm / sqrt(MeCo)**) to the best one in the final generation (Equation 2 in the manuscript).
> >
> > First, there is a noticeable increase in complexity, with the number of metrics rising from 3 (ZiCo, L2-norm, MeCo) to 6 (ZiCo, MeCo, Zen, SNIP, L2-norm, FLOPs). We suppose that this increased complexity enables the synthesized metric to capture more characteristics of architectures. Second, the core metrics ZiCo, L2-norm, and MeCo are consistently retained across both versions, demonstrating that the SR model effectively identifies and preserves metrics that contribute significantly to high performance. Lastly, the increased complexity results in substantial improvements in Kendall’s Tau scores, indicating a stronger correlation with true performance: from 0.39, 0.70, and 0.35 (for the initial synthesized metric) to 0.61, 0.76, and 0.40 (for the final synthesized metric) across three NAS problems.
> >
> > Lastly, we explore the impact of replacing the most frequently used ZC metrics in our synthesized metric (e.g., ZiCo, Snip, and MeCo in Equation 2) with the least frequently used ones (e.g., SWAP, Grasp, and Plain) and compare the Kendall’s Tau scores of these two variants. The new following results reveal that the effectiveness of the synthesized metric is significantly reduced in all replacement cases. This result, coupled with the presence of effective metrics like ZiCo and MeCo in both the initial and final populations, demonstrates that our SR framework effectively identifies these metrics as crucial components when combined with others to form potential “building blocks”. The SR framework then assembles these building blocks to create high-performing ZC metrics. Therefore, substituting components within these building blocks disrupts their structures, leading to a noticeable decline in the performance of the synthesized metric.
> > | Replacing |      NB101-CF10 |  NB201-CF10 |  NB301-CF10 |
> > |:----------|:-------------:|:------:|:------:|
> > | MeCo → SWAP | 0.52 | 0.41 | 0.39 |
> > | ZiCo → Grasp | 0.15 | 0.45 | 0.22 |
> > | Snip→ Plain  | 0.43 | 0.71 | 0.33 |
> > | **Ours (Equation 2)** | **0.61** | **0.76** | **0.40**|

---

> ### Author Response · Authors · 2024-11-23
>
> **Weaknesses**
> > 1. The innovation of the paper is limited, the framework is a common symbolic regression to search the best mathematical expressions of ZC metrics.
>
> Thank you for your thoughtful comment on the innovation of our work. While we acknowledge that Symbolic Regression (SR) itself is a widely used approach, we selected SR for several specific reasons that align with the goals of our study. These reasons, combined with the novel application of SR to the problem of synthesizing ZC metrics for diverse NAS problems, demonstrate the unique contributions of our framework:
>
> 1. **Suitability for Tabular Data**:
> The NAS-Bench-Suite-Zero database, which forms the basis of our study, consists of tabular data. In exploring methods to effectively utilize this rich and diverse dataset, we found that traditional machine learning methods, such as SR, are well-suited for such data. Deep learning (DL)-based approaches, while powerful in other contexts, may require more extensive data and resources and might not leverage the inherent structure of tabular data as effectively. Empirically, our approach demonstrates superior performance compared to UP-NAS [1], which employs a DL model to optimize weights between ZC metrics. Additional experiments conducted in the revised manuscript further support the effectiveness of our method compared to other approaches that combine multiple ZC metrics.
>
> 2. **Interpretability**:
> One significant advantage of SR is its ability to produce interpretable mathematical expressions. This human-readable representation allows us to derive valuable insights from the synthesized ZC metric. For instance, in a related work (AutoProx [2]), SR-based design of ZC metrics revealed that specific weight parameters in linear layers of MLP modules significantly influence performance.
>
> 3. **Comparisons with Related Work**:
> Two related studies, i.e., EZ-NAS [3] and AutoProx [2], also employ SR to design new ZC metrics. However, our work introduces key differences and advancements:
>     - **Input Features**: Our framework uses high-level ZC metrics as input features, offering greater extensibility compared to EZ-NAS and AutoProx, which rely on low-level features. High-level ZC metrics are more generalizable and transferable across tasks, a property critical for evaluating diverse NAS problems.
>     - **Objective Function**: The objective function in our framework evaluates the effectiveness of expression trees across multiple NAS problems without requiring predefined weights for each problem. In contrast, AutoProx requires such weights to be set beforehand, which introduces subjectivity and limits adaptability.
>
> In summary, while the SR framework itself is established, our contributions lie in (1) adapting it to the specific context of ZC metric design, (2) demonstrating its effectiveness in synthesizing interpretable and generalizable metrics, and (3) addressing limitations in related work through novel design choices in input features and evaluation functions. We hope this response clarifies the innovative aspects of our work and its contributions to the field of NAS and ZC metric design.
>
> ----------
>
> [1] Yi-Cheng Huang, Wei-Hua Li, Chih-Han Tsou, Jun-Cheng Chen, and Chu-Song Chen. UP-NAS: Unified Proxy for Neural Architecture Search. In IEEE Conference on Computer Vision and Pattern Recognition (CVPR) Workshops, 2024.
>
> [2] Zimian Wei, Peijie Dong, Zheng Hui, Anggeng Li, Lujun Li, Menglong Lu, Hengyue Pan, and Dongsheng Li. Auto-Prox: Training-Free Vision Transformer Architecture Search via Automatic Proxy Discovery. In The Conference on Artificial Intelligence (AAAI), 2024.
>
> [3] Yash Akhauri, Juan Pablo Munoz, Nilesh Jain, and Ravi Iyer. EZNAS: Evolving Zero-Cost Proxies For Neural Architecture Scoring. In Advances in Neural Information Processing Systems (NeurIPS), 2022.

---

> ### Author Response · Authors · 2024-11-23
>
> **Question**
> > 1. The Eq.1 is essentially a normalization, It also looks like giving each question the same weight.
>
> Thank you for your comment. You are correct that Eq. 1 serves as a normalization mechanism, and we appreciate the opportunity to clarify its purpose.
>
> In our approach, the objective function values of each candidate ZC metric during the symbolic regression search process are determined by Kendall’s Tau scores corresponding to multiple NAS problems. To deal with the difference in the best and worst Kendall’s Tau scores obtained for each problem, we apply normalization to scale these values.
>
> To illustrate this, consider two tasks, A and B, where Task A has the lowest and highest Kendall’s Tau scores of (0.2, 0.9), and Task B has scores of (0.1, 0.4). Now, if we compare two ZC metrics (ZC_1 and ZC_2) with their corresponding Kendall’s Tau values on Tasks A and B as follows:
>
> - **ZC_1**: (0.8, 0.1)
> - **ZC_2**: (0.5, 0.3)
>
> If we were to simply sum the raw Kendall’s Tau scores without any normalization (i.e., equal weighting), we would choose **ZC_1**, as it has the higher score (0.9 > 0.8). However, this approach would ignore the fact that **ZC_2** performs better across both tasks, while **ZC_1** excels only in Task A.
>
> When we apply normalization, the adjusted scores would be:
>
> - **ZC_1**: (0.86, 0.0)
> - **ZC_2**: (0.43, 0.67)
>
> In this case, **ZC_2** would be preferred, as its normalized scores sum to 1.1, which is higher than **ZC_1**'s sum of 0.86. This reflects our goal of selecting a ZC metric that performs well across diverse NAS problems, rather than excelling in only one.
>
> Thus, the normalization in Eq. 1 allows us to give appropriate weight to tasks based on their difficulty, ensuring that a metric’s performance is evaluated holistically across multiple tasks.
>
> We hope this clarifies your concern about Eq. 1 and the rationale behind using this normalization approach.

---

> ### Author Response · Authors · 2024-11-23
>
> **Questions**
> > 3. What is the multiple problems in dataset? Are they different tasks?
>
> We appreciate the opportunity to clarify.
>
> As we provided in footnote 2 on page 2 of the paper, **an NAS problem** refers to a combination of a **single search space** and a **single specific task**. For example, DARTS and NAS-Bench-101 are two distinct search spaces. Performing image classification on CIFAR-10 and performing image classification on ImageNet are two distinct tasks.
>
> In our approach, the input dataset for Symbolic Regression covers *multiple problems* and it thus contains multiple search spaces *SS = {SS_1, SS_2, …, SS_i}* and tasks *T = {T_1, T_2, …, T_i}*. We note that two distinct problems could not have the same search space and task. They could have the same search space (e.g., NAS-Bench-101) but would have different tasks (e.g., image classification on CIFAR-10, image classification on ImageNet) and *vice versa*. In our main experiments (Section 4.1), the dataset used to search for the new ZC metric consists of three NAS problems that have different search spaces (i.e., NAS-Bench-101, NAS-Bench-201, NAS-Bench-301) but the same task (image classification on CIFAR-10 dataset). Besides, we also experimented with the dataset containing the problems having different tasks (Image classification on CIFAR-10 dataset and Scene classification on Taskonomy dataset) in the ablation study (Section 4.4).
>
> We hope this clarifies your concern.
>
> > 4. There seems to be a typo in the annotation of Figure 2. NWOT should be MeCo.
>
> Thank you for pointing out this oversight. We will correct all typos in the revised manuscript.
>
> > 5. In Figure 4, what do SS^i and T^j in the data set represent?
>
> Thank you for pointing out this missing. As responded in Question #3, our dataset consists of multiple NAS problems and it thus contains multiple search spaces *SS = {SS_1, SS_2, …, SS_i}* and tasks *T = {T_1, T_2, …, T_i}*. *SS_i* refers to the *i*-th search space in the list of search spaces *SS* and *T_j* refers to the *j*-th in the list of tasks *T*. We will clarify this information in the caption of Figure 4 in the revised manuscript.

---

> ### Author Response · Authors · 2024-11-24
>
> **Questions**
> > 2. The paper focuses on the performance of the designed ZC metrics but does not provide insights into why certain combinations of metrics work better than others.
>
> In response to Weakness #2, we have conducted an analysis of the resulting synthesized ZC metrics and the search process of our proposed SR framework and we will attach the insights from the analyses to the revised manuscript.
>
> When comparing the best expression tree in the first generation (i.e., **ZiCo * L2-norm / sqrt(MeCo)**) to the best one in the final generation (Equation 2 in the manuscript), we obtain some interesting findings.
>
> First, there is a noticeable increase in complexity, with the number of metrics rising from 3 (ZiCo, L2-norm, MeCo) to 6 (ZiCo, MeCo, Zen, SNIP, L2-norm, FLOPs). We suppose that this increased complexity enables the synthesized metric to capture more characteristics of architectures. Second, the core metrics ZiCo, L2-norm, and MeCo are consistently retained across both versions, demonstrating that the SR model effectively identifies and preserves metrics that contribute significantly to high performance. Lastly, the increased complexity results in substantial improvements in Kendall’s Tau scores, indicating a stronger correlation with true performance: from 0.39, 0.70, and 0.35 (for the initial synthesized metric) to 0.61, 0.76, and 0.40 (for the final synthesized metric) across three NAS problems.
>
> Lastly, we explore the impact of replacing the most frequently used ZC metrics in our synthesized metric (e.g., ZiCo, Snip, and MeCo in Equation 2) with the least frequently used ones (e.g., SWAP, Grasp, and Plain) and compare the Kendall’s Tau scores of these two variants. The new following results reveal that the effectiveness of the synthesized metric is significantly reduced in all replacement cases. This result, coupled with the presence of effective metrics like ZiCo and MeCo in both the initial and final populations, demonstrates that our SR framework effectively identifies these metrics as crucial components when combined with others to form potential “building blocks”. The SR framework then assembles these building blocks to create high-performing ZC metrics. Therefore, substituting components within these building blocks disrupts their structures, leading to a noticeable decline in the performance of the synthesized metric.
> | Replacing |      NB101-CF10 |  NB201-CF10 |  NB301-CF10 |
> |:----------|:-------------:|:------:|:------:|
> | MeCo → SWAP | 0.52 | 0.41 | 0.39 |
> | ZiCo → Grasp | 0.15 | 0.45 | 0.22 |
> | Snip→ Plain  | 0.43 | 0.71 | 0.33 |
> | **Ours (Equation 2)** | **0.61** | **0.76** | **0.40**|
>
> We hope this response clarifies all your concerns.

---

### Author Response · Authors · 2024-11-25
**Summary of updates in the revised manuscript**

We sincerely thank reviewers yYgB, c1zS, and ubay for their insightful comments. Your valuable feedback has significantly improved our manuscript. In response, we have made the following adjustments in the revised version:

- Corrected the typo in the caption of Figure 2.
- Clarified the definitions of SS_i​ and T_j ​ in the caption of Figure 4.
- Based on the collective suggestion to evaluate the performance on ImageNet, we conducted additional experiments within the Once-For-All search space and included the new results in Section 4.3.
- To address Reviewer ubay's concern regarding the comparison of our Symbolic Regression framework with related works such as AZ-NAS and RoBoT, we conducted a performance comparison and added the results in Appendix J.
- In response to Reviewer yYgB's comments, we have included valuable insights in Appendix K from analyzing the synthesized ZC metric and the search process of our Symbolic Regression framework.

We greatly appreciate the reviewers' constructive feedback, which has helped us enhance the quality and the scope of our work.

---

### Meta-Review · Area_Chair_SSyi · 2024-12-23

**Metareview:**

This paper proposes a novel framework utilizing symbolic regression for the automatic synthesis of zero-cost (ZC) proxy metrics for neural architecture search (NAS). The reviewers acknowledged the relevance of the problem and the potential of the framework to streamline ZC metric design. However, concerns were raised about the limited novelty, the framework’s performance in specific benchmarks, and the lack of clear theoretical insights into why certain combinations of metrics outperform others. Despite the authors’ rebuttal addressing several points, such as adding ImageNet results and clarifying the framework’s contributions, these efforts only partially alleviated the concerns. With a majority of the reviewers rating the paper below the acceptance threshold, the consensus leans towards rejection.

**Additional Comments On Reviewer Discussion:**

The discussion period emphasized three primary concerns: limited innovation compared to prior work, the need for validation on large-scale datasets like ImageNet, and the framework’s dependency on existing ZC metrics. The authors conducted additional experiments, including evaluations on the ImageNet dataset, and provided comparisons with related methods such as AZ-NAS and RoBoT. They also clarified their use of symbolic regression and addressed misunderstandings about the reported computational efficiency. While these updates demonstrated the approach's applicability and addressed specific points, the reviewers remained unconvinced about the framework’s generalizability and its incremental contribution to the field. These factors collectively influenced the final recommendation.

---

### Decision · Program_Chairs · 2025-01-22

Reject